# Emission factors and evolution of SO₂ measured from biomass burning in wild and agricultural fires

Pamela S. Rickly[1,2], Hongyu Guo[1,3], Pedro Campuzano-Jost[1,3], Jose L. Jimenez[1,3], Glenn M. Wolfe[4], Ryan Bennett[5], Ilann Bourgeois[1,2], John D. Crounse[6], Jack E. Dibb[7], Joshua P. DiGangi[8], Glenn S. Diskin[8], Maximilian Dollner[9], Emily M. Gargulinski[16], Samuel R. Hall[10], Hannah S. Halliday[11], Thomas F. Hanisco[4], Reem A. Hannun[4,12], Jin Liao[4,13], Richard Moore[8], Benjamin A. Nault[14], John B. Nowak[8], Jeff Peischl[1,2], Claire E. Robinson[8,15], Thomas Ryerson[2,a], Kevin J. Sanchez[8], Manuel Schöberl[9], Amber J. Soja[8,16], Jason M. St. Clair[4,12], Kenneth L. Thornhill[8], Kirk Ullmann[10], Paul O. Wennberg[6,17], Bernadett Weinzierl[9], Elizabeth B. Wiggins[8], Edward L. Winstead[8], and Andrew W. Rollins[2]

[1]Cooperative Institute for Research in Environmental Science, University of Colorado, Boulder, CO, USA

[2]Chemical Sciences Laboratory, NOAA, Boulder, CO, USA

[3]Department of Chemistry, University of Colorado, Boulder, CO, USA

[4]Atmospheric Chemistry and Dynamics Lab, NASA Goddard Space Flight Center, Greenbelt, MD, USA

[5]Bay Area Environmental Research Institute, NASA Ames Research Center, Moffett Field, CA

[6]Division of Geological and Planetary Sciences, California Institute of Technology, Pasadena, CA, USA

[7]Earth System Research Center, University of New Hampshire, Durham, NH, USA

[8]NASA Langley Research Center, Hampton, VA, USA

[9]Faculty of Physics, Aerosol Physics and Environmental Physics, University of Vienna, 1090 Vienna, Austria

[10]Atmospheric Chemistry Observations and Modeling Laboratory, National Center for Atmospheric Research, Boulder, CO, USA

[11]Environmental Protection Agency, Research Triangle, NC, USA

[12]Joint Center for Earth Systems Technology, University of Maryland Baltimore County, Baltimore, MD 21250, USA

[13]Goddard Earth Science Technology and Research (GESTAR) II, University of Maryland Baltimore County, Baltimore, MD, USA

[14]CACC, Aerodyne Research, Inc.

[15]Science Systems and Applications, Inc., Hampton, VA, USA

[16]National Institute of Aerospace, Resident at NASA Langley Research Center, Hampton, VA,
USA

[17]Division of Engineering and Applied Science, California Institute of Technology, Pasadena,
CA, USA

[a]now at: Scientific Aviation, Boulder, CO, USA

*Correspondence to:* Pamela S. Rickly (pamela.rickly@state.co.us) and Andrew W. Rollins
(andrew.rollins@noaa.gov)

**Abstract.** Fires emit sufficient sulfur to affect local and regional air quality and climate. This
study analyzes $SO_2$ emission factors and variability in smoke plumes from U.S. wild and
agricultural fires, and their relationship to sulfate and hydroxymethanesulfonate (HMS)
formation. Observed $SO_2$ emission factors for various fuel types show good agreement with the
latest reviews of biomass burning emission factors, producing an emission factor range of 0.47 –
1.2 g $SO_2$ $kg^{-1}$ C. These emission factors vary with geographic location in a way that suggests
that deposition of coal burning emissions and application of sulfur-containing fertilizers likely
play a role in the larger observed values, which are primarily associated with agricultural
burning. A 0-D box model generally reproduces the observed trends of $SO_2$ and total sulfate
(inorganic + organic) in aging wildfire plumes. In many cases, modeled HMS is consistent with
the observed organosulfur concentrations. However, a comparison of observed organosulfur and
modeled HMS suggests that multiple organosulfur compounds are likely responsible for the
observations, but that the chemistry of these compounds yield similar production and loss rates
to that of HMS, resulting in good agreement with the modeled results. We provide suggestions
for constraining the organosulfur compounds observed during these flights and we show that the
chemistry of HMS can allow for organosulfur to act as a S(IV) reservoir under conditions of pH
> 6 and liquid water content > $10^{-7}$ g $sm^{-3}$. This can facilitate long-range transport of sulfur
emissions resulting in increased $SO_2$ and eventually sulfate in transported smoke.

**1 Introduction**

Sulfate is a major component of $PM_{2.5}$, contributing significantly to adverse air quality and
severe haze events (Chan and Yao, 2008). A severe haze event in Beijing, China showed $PM_{2.5}$
sulfur concentrations reaching 100 μg $m^{-3}$ with aerosol optical depths over 1 (Moch et al., 2018).
Sulfate aerosols are produced through the oxidation of sulfur dioxide ($SO_2$) which was estimated
to have a global emission rate of approximately 113 Tg S $yr^{-1}$ in 2014 (Hoesly et al., 2018).
Approximately 67% of global $SO_2$ emissions are due to anthropogenic sources, primarily fossil
fuel combustion and smelting (Lee et al., 2011; Smith et al., 2011; Feinberg et al., 2019).
       While biomass burning is expected to contribute a smaller portion to global sulfur
emissions (1.22 Tg S $yr^{-1}$), the effects of climate change and land use change are expected to
increase biomass burning events in both frequency and duration (Westerling et al., 2006;

Heyerdahl et al., 2002; Lee et al., 2011). Biomass burning $SO_2$ emissions can influence air
quality through sulfate aerosol production in regions thousands of kilometers away from the burn
site due to meteorological long-range transport (Fiedler et al., 2011). In extreme cases, pyro-
cumulonimbus formation injects biomass burning aerosol – including sulfate – into the upper
troposphere and lower stratosphere (Fromm et al., 2005).

Biomass burning produces both primary and secondary aerosols, with sulfate aerosols
resulting mostly from secondary production, but with a smaller primary component in some
cases (Lewis et al., 2009). The chemical composition of aerosols produced during biomass
burning is highly dependent on the environmental conditions and type of combustion occurring,
flaming or smoldering. For example, elemental carbon and $NO_x$ are mainly emitted during the
flaming stage, while emissions of VOCs and (mainly organic) $PM_{2.5}$ are larger during the
smoldering phase (Pandis et al., 1995; Lobert et al., 1991; Burling et al., 2010). Fuel composition
also influences $SO_2$ emissions. This is demonstrated in a recently published compilation of
biomass burning emission factors utilizing only data from young smoke to limit conversion
during chemical aging, reducing the variability within the published measurements (Andreae,
2019). This compilation shows savanna and grassland $SO_2$ emission factors to be $0.47 \pm 0.44 SO_2$
$kg^{-1}$ C and those for agricultural residues to be $0.80 \pm 0.71$ g $SO_2$ $kg^{-1}$ C with a full fuel type
range of 0.2 to 0.87 g $SO_2$ $kg^{-1}$ C.

Oxidation of $SO_2$ in both the gas and aqueous phase produces sulfate, with a typical $SO_2$
lifetime of 0.6 – 2.6 days (Pham et al., 1995; Koch et al., 1999). However, the importance of
some conversion mechanisms of $SO_2$ to sulfate remains poorly understood, resulting in the
frequent underprediction of sulfate concentrations by up to a factor of two for regional
atmospheric models (Wang et al., 2016; Shao et al., 2019; Wang et al., 2014). This
underprediction has been reported for industrialized pollution where limited photochemistry is
observed as a result of aerosol dimming (Cheng et al., 2016; Shao et al., 2019). While no known
studies have reported on the modeling of $SO_2$ and sulfate chemistry in biomass burning smoke
plumes, it is possible that similar phenomenon could occur because biomass burning plumes can
have very high aerosol loading and thus dimming. However, the chemistry is likely to be
different as a result of differing emissions. In addition, it has been suggested that unaccounted-
for hydroxymethanesulfonate (HMS) formation may explain the discrepancy between measured
and modeled sulfate values (Dovrou et al., 2019; Song et al., 2021).
In this study, we quantify $SO_2$ emissions and examine the production of sulfate using
airborne observations within a variety of smoke plumes. These measurements provide insight
into the variable emission factors observed during biomass burning and allow for a
comprehensive analysis of the conversion of $SO_2$ to sulfate and HMS including both gas- and
aqueous-phase conditions. Smoke is a highly dynamic environment, and we examine howsulfur
chemistry is affected by radiation attenuation, enhanced aerosol liquid water content (LWC), and
variable pH.

## 2 Methods

### 2.1 Mission and measurements

FIREX-AQ was a joint NASA-NOAA mission to study multiple aspects of fire emissions,
chemistry, and impacts. Here we utilize observations from the NASA DC-8. The base locations
for this aircraft campaign were Boise, ID, from 21 July to 17 August and Salina, KS, from 18

August to 5 September, 2019. The Boise location allowed for the measurement of western U.S. wildfires, with sampling occurring in the late afternoon through evening. Salina-based flights focused on prescribed burns, primarily of croplands, within the midwestern and southern regions of the U.S. with measurements typically occurring in the afternoon. A subset of these measurements including seven different fuel types from over 80 fires is reported here.

Flight paths differed between the wildfire and cropland measurements. A typical flight path through the wildfire smoke plumes consisted of two "lawnmower" patterned passes consisting of about 10 staggered downwind transects perpendicular to the plume (Fig. 1). The closest transects were generally 10—15 km downwind due to flight restrictions, with the pattern extending as far as 200 km downwind, resulting in smoke ages (based on Lagrangian trajectory

analysis) ranging from tens of minutes to several hours. In contrast, sampling of smaller agricultural fires typically involved 1—2 plume transects per fire.

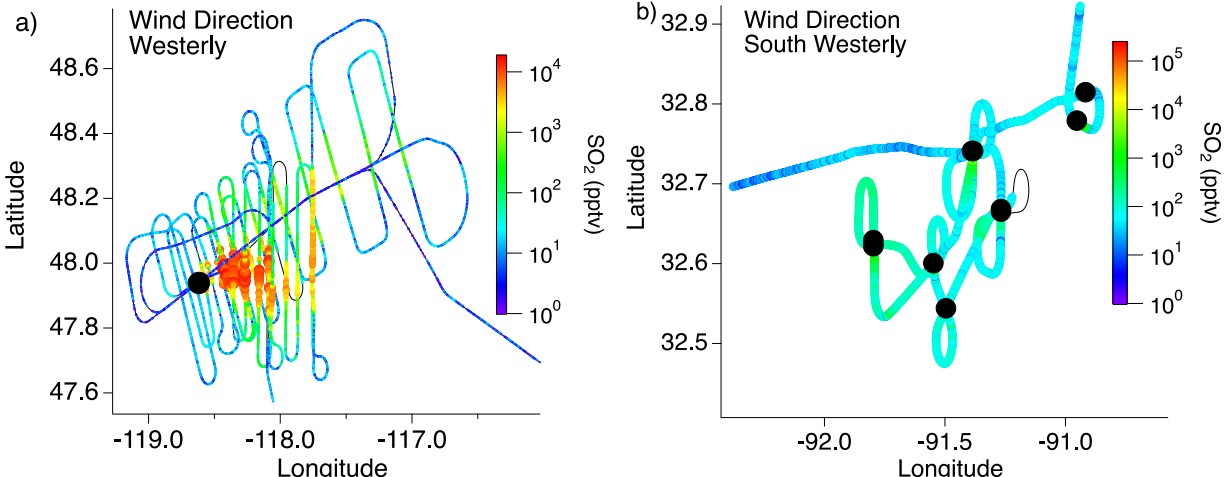

*Figure 1. Typical flight path through (a) wildfire and (b) agricultural fire smoke plumes with the*
*color and size of the markers indicating the SO$_2$ mixing ratio and the black markers indicating the fire locations.*

In situ measurements of SO$_2$ were performed using laser induced fluorescence (LIF SO$_2$) in which SO$_2$ was excited at 216.9 nm by a custom-built fiber laser system with the red-shifted

fluorescence detected between 240 and 400 nm. An intercomparison performed between the LIF SO$_2$ and Caltech CIMS instrument during FIREX-AQ showed good agreement between the two measurement techniques (Rickly et al., 2021). The accuracy of the LIF SO$_2$ measurements is ±9% + 2 pptv, primarily dictated by uncertainty in the calibration standard concentration and spectroscopic background.

Sulfate measurements were performed by a suite of in-situ instruments: an Aerodyne high-resolution time-of-flight aerosol mass spectrometer (AMS) (DeCarlo et al., 2006; Canagaratna et al., 2007), with a sampling rate of 1-5 Hz, the online soluble acidic gases and aerosol mist chamber (SAGA-MC) coupled with ion chromotograph (IC) (Scheuer et al, 2003; Dibb et al, 2003), with a sampling interval of 75 s) and SAGA filter collector with subsequent

offline IC analysis (Dibb et al., 1999; Dibb et al., 2000), with tyical sampling intervals of 3 min in the large fires. Both SAGA-MC and AMS sample submicron particles, while the SAGA filter collects both submicron and supermicron particles up to 4.1 μm with 50% transmission (McNaughton et al., 2007; van Donkelaar et al., 2008; Guo et al., 2021). The AMS instrument

allows for the speciation of submicron non-refractory particulate mass and the direct separation of inorganic and organic species having the same nominal mass to charge ratio (DeCarlo et al., 2006; Canagaratna et al., 2007). Both inorganic and organic sulfate fragment similarly in the AMS, mostly to $H_xSO_y^+$ ions without carbon. For AMS total nitrate, where the fragmentation pattern is similar (Farmer et al, 2010), , techniques for rapid assignment of organic nitrate based on its fragmentation pattern have been successfully developed (Fry et al., 2013; Day et al., 2021). While there are some differences in fragmentation between organic and inorganic sulfur that have been used in some cases to separate organic from inorganic sulfate (Chen et al., 2019; Dovrou et al., 2019); the sulfate fragmentation pattern is overall much more variable compared to nitrate and hence such approaches will work only in very specific instances (Schueneman et al., 2021). In this work, we found the ion fragmentation method to produce reasonable results, based on the consistency with the results using positive matrix factorization (PMF, Paatero et al., 1994, Ulbrich et al., 2009) and the measurements of submicron sulfate aerosol from SAGA-MC, which quantifies only inorganic sulfate. The correlation between the AMS inorganic sulfate and SAGA-MC sulfate shows an overall good agreement (Fig. S8), which adds confidence to the AMS apportionment. However, as discussed in section 4.2.2, for certain types of organosulfur compounds, hydrolysis in the liquid phase after capture into the instrument and before analysis might lead to SAGA-MC detecting these as well, hence the SAGA-MC sulfate measurements are likely more uncertain under FIREX-AQ conditions based on the default accuracy estimates for this instrument (Dibb et al., 2002; Scheuer et al., 2003).

Both IC (SAGA) instruments detect HMS as S(IV), and the signal interfered with sulfite and bisulfite. There is no unambiguous detection of HMS specifically, either in the IC or in the AMS.

In situ CO concentrations were measured via wavelength modulation spectroscopy (Sachse et al., 1991), with an uncertainty of 2—7% over the dynamic range of the measurements. In situ $CO_2$ concentrations were measured using non-dispersive infrared spectrometry using a modified commercial spectrometer (Model 7000, LI-COR) similar to Vay et al. (2009), with uncertainties varying between 0.25 ppm and 2% of the measurements (whichever is larger) over the range of the measurements.

## 2.2 Emission factor calculation

Emission factors (EF) are defined as the mass of compound X relative to the mass of fuel burned; however, this can be substituted with the mass balance method which approximates the fuel mass by the sum of emitted carbon (Andreae, 2019).  In accordance, the emission factors for $SO_2$ and sulfate were calculated as the enhancement ratio of each compound relative to the enhancement ratio of total carbon emitted per fire in units of g kg$^{-1}$ (Eq. 3.1). Because CO and $CO_2$ comprise approximately 95% of total carbon emissions, the summation of these values was used to represent total carbon.

$$EF(X) = \frac{X}{CO+CO_2} \cdot \frac{MM_X}{MM_C} \cdot F_C \cdot 1000 \qquad (3.1)$$

The orthogonal distance regression slope of compound X to total carbon ($\frac{X}{CO+CO_2}$) was determined for each transect through the smoke plume with a smoke age < 1 hr to limit the influence of chemical processing due to atmospheric aging. Only emission ratio values with $R^2 >$

0.5 were included in the EF analysis. It is shown in sections 3.3 and 3.7 that no significant aging
of $SO_2$ occurs within this length of time. In addition, only measurements $\geq 25\%$ enhanced from
the background were used, which allowed for the background mixing ratios to be neglected.
$MM_X$ and $MM_C$ represent the molar mass of compound X and the summation of CO and $CO_2$,
respectively. The approximated value of 45% is used to represent the carbon fraction ($F_C$) of the
fuel emitted during these biomass burning events as outlined by Susott et al. (1996) and allows
for a more direct comparison to the compilation of EF data prepared by Andreae (2019).

## 2.3 Modified combustion efficiency

The modified combustion efficiency (MCE) is a metric for combustion stage. The MCE is
defined as the enhancement of $CO_2$ from the background in relation to the summation of the
enhanced CO and $CO_2$ mixing ratios (Eq. 3.2). Traditionally, MCE > 0.9 is indicative of the
flaming stage and an MCE < 0.9 is representative of the smoldering stage (Ferek et al., 1998;
Sinha et al., 2003; Zhang et al. 2018). In reality, smoke sampled from large wildfires likely
reflects a combination of variable fractions of flaming and smoldering combustion.

$$MCE = \frac{CO_2}{CO + CO_2} \qquad (3.2)$$

## 2.4 Box Model

The Framework for 0-D Atmospheric Modeling (F0AM, Wolfe et al., 2016) was used to evaluate
the evolution of $SO_2$ downwind of the fire location (Wolfe et al., 2016). Within F0AM, the
Master Chemical Mechanism (MCM) version 3.3.1 was used to describe the evolution and
chemistry of the gas-phase $SO_2$ and oxidant species. An additional mechanism describing the
conversion of $SO_2$ to sulfate was implemented to address aerosol oxidation processes of sulfur
compounds based on an establishment of equilibrium of the S(IV) compounds and oxidant
species with relation to pH (Tang et al., 2014; D'Ambro et al., 2016; Seinfeld and Pandis, 2006).
A complete list of the aqueous phase reactions and measurements used for model input is
included in Tables S1 & & S2 and the mechanism code is provided in the Supplementary Section
2.
        The model was implemented to investigate the chemistry that occurred during the
Williams Flats fire which started 2 August 2019by lightning ignition of timber/slash fuels in
Keller, WA. Two separate flight days, 3 and 7 August, were modeled here using measurements
acquired by the DC-8 in which two passes of lawnmower patternspatternswere completed. These
flights were analyzed by applying a Lagrangian model approach. The measurements were
corrected for dilution by normalizing to CO (Müller et al., 2016) as follows:

$$\Delta_{dil}X = \frac{(X - X_b)}{(CO - CO_b)} \cdot CO_i \qquad (3.3)$$

in which $\frac{X}{CO}$ represents the ratio of compound X at each transect with respect to CO, $X_b$ and $CO_b$
are the background concentrations, and $CO_i$ represents the carbon monoxide mixing ratio at the
source of the fire determined from the extrapolation of the transect average CO values. This
extrapolation method was also applied to the dilution-normalized mixing ratios in order to

initialize the model back to the fire source (t=0). The model was constrained to these initial
concentrations, then allowed to run freely through the remainder of the flight time. The dilution
rate was determined by matching the modeled CO to the measured CO decay using a Gaussian
fit. However, $CO_i$, used to determine thedilution-normalized mixing ratio values, was based on
the extrapolated CO initial value based on all transect CO values (core and edge).

Measurements were acquired through aircraft smoke plume penetration, which provided
pseudo-Lagrangian observations by not entirely following the same air parcel. Comparison to a
Lagrangian simulation is challenging because the aircraft measured different parts of the plume
(core vs. edge) and at different emission times. As a result, an exponential fit applied to the $SO_2$
and sulfate dilution-normalized mixing ratios against plume age is used to represent the
measurement trend for comparison to the model results. While the model is not expected to
precisely reproduce the measurements based on plume age due to variations in altitude between
transects and subsequently varied pressures and temperatures, it does allow for the comparison of
the overall trends of $SO_2$ and sulfate downwind of the source using averaged meteorological
constraints.

Uptake of $SO_2$ and the oxidant species ($O_3$, $NO_2$, $H_2O_2$, and HCHO) to aerosol was
represented within the model mechanism as a first-order loss (Seinfeld and Pandis, 2006):

$$khet = 0.25 \cdot \gamma \cdot c \cdot \nu \tag{3.4}$$

where $\gamma$ represents the uptake coefficient, c is the mean molecular speed of $SO_2$, and $\nu$ is the
aerosol surface area based on average dry particle size distributions measured by a Laser Aerosol
Spectrometer 3340. To account for the gas-phase diffusion limitation, $\gamma$ was calculated by the
following equation:

$$\gamma = \frac{1}{\alpha} + \frac{0.75 + 0.286 Kn^{-1}}{Kn \cdot (Kn+1)} \tag{3.5}$$

where $\alpha$ represents the mass accommodation coefficient and Kn is the Knudsen number. Mass
accommodation and gas diffusion coefficients used for deriving Kn and $\gamma$ are listed in Table S3.

To represent equilibrium partitioning between the gas and aqueous phases, rates of
condensation and evaporation were applied as described by D'Ambro et al. (2016):

$$k_{cond} = khet \tag{3.6}$$

$$k_{evap} = \frac{khet}{H \cdot LWC} \tag{3.7}$$

where H represents the Henry's Law constant of the species being adsorbed and LWC is the
liquid water content of the cloud or aerosol. The dry particle size (not ambient particle size) is
incorporated into khet through Eq. 3.4. This khet value is then applied to Eq. 3.7 as a ratio to the
LWC and ability of uptake (H), allowing for calculation of the gas-particle equilibrium.
Therefore, as the particle size increases, greater condensation is able to occur, but this also
allows for increased evaporation. However, with an increase in LWC and H, less evaporation
will be expected. Using this method of uptake and evaporation does not allow for equilibrium of
all processes to be assumed as isis done in the ISORROPIA calculations. Because S(IV)
production is pH dependent, individual equilibrium constants in relation to the $H^+$ produced by
each reaction are required as an additional factor in the $k_{evap}$ denominator as described by

Seinfeld and Pandis (2006). As a result, the model accurately reproduces the S(IV) pH
       dependence (Fig. S1a) in which $HSO_3^-$ is the dominant form between the pH range of 2—7 and
       $SO_3^{2-}$ becomes the dominant form at pH > 7. Table S1 lists all aqueous phase reactions.
             The rate of S(IV) oxidation exhibits a pH dependence based on the available oxidant
       species (Table S1) (Cheng et al., 2016). Using our model and the initial conditions from Guo et
al. (2017), we reproduced very similar pH dependent oxidation rates to those shown in that
       study. However, initializing the model with the higher concentrations observed during FIREX-
       AQ increases the rates of oxidation as shown in Fig. S1b. This results in S(IV) oxidation being
       dominated by reaction with hydrogen peroxide at pH values < 5 which is within the range that
       aerosol sulfate production most commonly occurs in the U.S. For pH values approaching 5, there
may be some competition amongst $H_2O_2$, $O_3$, and HCHO depending on the oxidant
       concentrations. As pH values increase above 5, $O_3$, $NO_2$, and HCHO become the dominant
       oxidants with $H_2O_2$ and $NO_2$ oxidation declining rapidly. Although the reaction of HCHO with
       S(IV) results in HMS production rather than inorganic sulfate, it has been included here to
       demonstrate its impactimpact on S(IV) oxidation. HCHOadduct formation follows a very similar
trend to $O_3$ oxidation, becoming a major S(IV) reactant at higher pH. Further discussion of the
       HMS reactions listed in Table S1 can be found in the supplement.
             In this study, aerosol LWC and pH were determined via ISORROPIA-II thermodynamic
       modeling (Fountoukis and Nenes, 2007) in forward mode based on the AMS measured aerosol
       composition ($SO_4$, $NO_3$, $NH_4$, Cl) and collocated gas-phase measurements of $NH_3$ and $HNO_3$
from PTR-MS and CIMS, respectively. $NH_3$-$NH_4$ is the most important species pair for
       constraining pH because it was not completely in either the gas or particle phase in the fire
       plumes or the background air mass. To improve the accuracy in thermodynamic modeling
       predictions, we removed the outliers when the predicted particle phase fraction of the $NH_3$-$NH_4$
       partitioning is off by > 40% compared to the observation (4.6% of the data). The gas-particle
partitioning is reproduced with ISORROPIA-II, with the regression slopes of predicted $NH_3$,
       $NH_4$, and $NO_3$ close to one compared to the observations and highly correlated (slopes: 0.949,
       1.116, and 1.002; $r^2$: 0.991, 0.96, and 0.99996, respectively). This also supports the assumption
       of equilibrium, as the characteristic time for fine particle water equilibrium is very short (< 1 s)
       (Pilinis et al., 1989) and ranges from 20 mins or less (Dassios and Pandis, 1999; Cruz et al.,
2000; Fountoukis et al., 2009; Guo et al., 2018) up to 10 hrs for semivolatile components, $NH_3$,
       $HNO_3$, and HCl (Meng and Seinfeld, 1996; Fridlind and Jacobson, 2000; Shingler et al., 2016).
       The uncertainty in particle pH is estimated to be within 0.5-1 unit based on the sensitivity of pH
       to $NH_3$-$NH_4$ partitioning and varies from point to point depending on the model reproduction of
       the partitioning (Guo et al., 2017). Because these calculations are based on the inorganic aerosol
concentrations, the LWC could potentially be up to several times greater due to the dominant
       organic portion in the fire plumes despite the lower hygroscopicity compared to the inorganics
       (Kreidenweis et al., 2008; Guo et al, 2015; Brock et al, 2016). The mixing state of inorganic and
       organic for the particles in the early phase plumes remains to be investigated but is likely to be
       phase separated given the low oxidation state of the organics (Sullivan et al., 2020). The current
modeling can be interpreted as assuming a phase separation of inorganic vs. organics, with the
       chemistry studied occurring only in the inorganic-dominated phase and its associated water, with
       no kinetic limitations due to potential core/shell ormicelle-like structures present in the particles.
       Propagating the uncertainties of AMS inorganics (34%, 2σ) (Bahreini et al, 2009)%) and DC-8
       totaltotal water measurement (3% based on the observed RH) gives an LWC uncertainty of 39%
(Guo et al., 2015). Due to the dominant organic fraction of sulfate signals in the fire plumes

investigated in this study, additionaladditional bias and uncertainty derive from using the total AMS $SO_4$ signals and zero non-volatile cations (e.g., not accounting for the potential contribution of soluble ions from ash, Adachi et al, 2022) in estimating LWC and pH. This is of particular concern when the uncertainties are larger than the estimated free acidity based on ion balance, as often happens near the neutralization point. The potential bias is estimated to be -0.96±0.95 unit for pH (i.e., biased low).

Most importantly, the modeling work presented in this study assumes an ideal solution. Given the relatively high ionic strength conditions observed for the 3 Aug (89.5 ± 19.3 M) and 7 Aug (83.2 ± 25.3 M) flights due to the overall rather low RH, this can potentially lead to high deviations in the actual gas uptake coefficients, aqueous phase rate coefficients and to a lesser extent, pH (calculation of which does account for ionic strength, but is fairly under constrained under these conditions).

## 3 Results and discussion

### 3.1 Emission factors

The elemental sulfur EFs calculated for FIREX-AQ are comparable to previous reports. As described in section 2.33, flaming and smoldering delineation was determined by an MCE value of 0.9. For consistency with other FIREX-AQ reports, the fuel types listed remain as subcategories, but are combined for comparison to the comprehensive biomass burning fuel types listed by Andreae (2019). The FIREX-AQ agriculture category comprises measurements of residual burns of rice, corn, and soybean fields. Across the fuel types measured during FIREX-AQ (Fig. 2), we find that $SO_2$ is consistently larger than sulfate when calculated as EFs of elemental sulfur, indicating that, at most, a minor fraction of $SO_2$ (20—25%) is converted to sulfate within 1 hr downwind (or emitted directly as primary sulfate). Where dataare not reported, this is due to either missing data or a low correlation with total carbon ($R^2 < 0.5$). The total sulfur EFs agree reasonably well with those reported by Andreae et al. (1988), measured in the Amazon basin, in the range of 0.24—0.66 g S $kg^{-1}$ C.

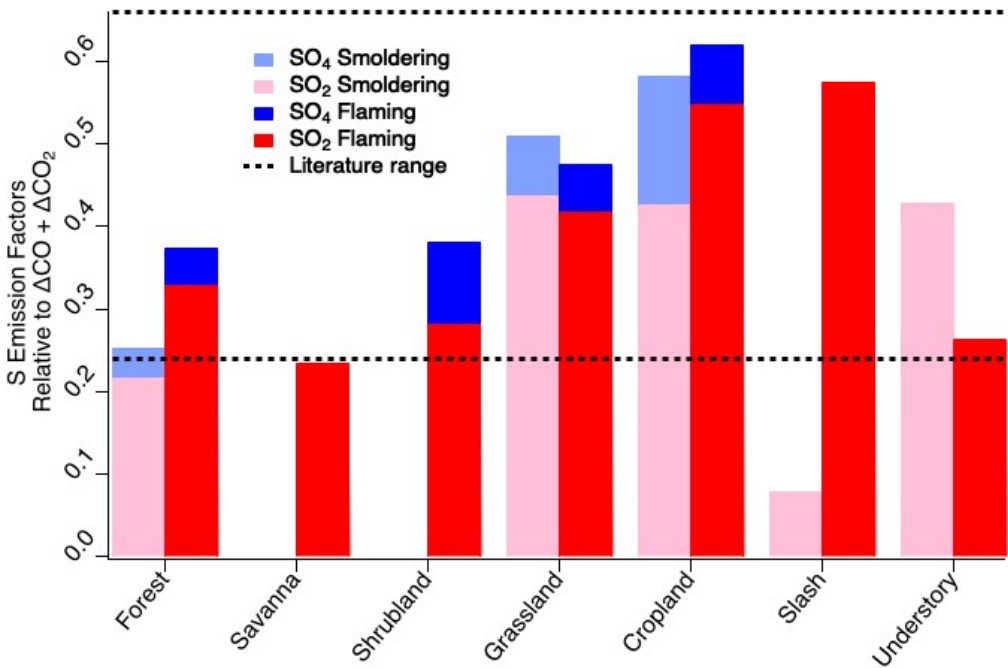

*Figure 2.* Elemental sulfur *emission factors of SO₂ and sulfate by fuel type and combustion stage within 1 hr downwind compared to literature values of total sulfur emission factors.*

No trend with MCE is observed for $SO_2$ EFs when separated by the various fuel types for smoldering and flaming conditions above MCE 0.85 for $SO_2$ and sulfate (Fig. 3). It has previously been suggested that EFs can be calculated based on MCE for use by the global climate modeling community. There have been conflicting opinions around this suggestion with some species showing relevant correlations while other species do not (Yokelson et al., 1996; Burling et al., 2011; Akagi et al., 2013). Considering all the EFs for $SO_2$, sulfate, and the ratio of $SO_2$ to sulfate under one hour shows that, individually, $SO_2$ and sulfate do not show strong correlations with MCE (Fig. 3). However, the ratio of the two produces a stronger correlation suggesting there may be a relationship in which more sulfate may be produced during smoldering combustion and more $SO_2$ emitted during flaming combustion. One possibility is that the smoke plumes from smoldering fires are more conducive to rapid conversion of $SO_2$ to sulfate such that the ratio of $SO_2$ and sulfate has significantly decreased by the time it is sampled. This could be due to a number of factors, including higher aerosol EF which, depending on the aerosol composition, could allow for more rapid aqueous phase oxidation. It is also possible that more primary sulfate is emitted from those plumes.

Averaging the flaming and smoldering EFs produces an overall $SO_2$ EF of $0.73 \pm 0.43$ g $SO_2$ $kg^{-1}$ C. This is within the combined variability of the Andreae (2019) compilation of flaming and smoldering EFs of $0.62 \pm 0.75$ g $kg^{-1}$ C, which excludes peat and laboratory fires. Separating the $SO_2$ EFs by combustion stage results in a flaming stage value of $0.80 \pm 0.46$ g $kg^{-1}$ C ($0.62 \pm 0.61$ g $kg^{-1}$ C from Andreae, 2019) and a smoldering stage value of $0.62 \pm 0.36$ g $kg^{-1}$ C ($0.61 \pm 0.27$ g $kg^{-1}$ C from Andreae, 2019). While the FIREX-AQ flaming stage value is considerably higher than the Andreae (2019) compilation, the two are within the combined variability of the observations. However, this higher average EF for the flaming stage FIREX-AQ measurements

is strongly influenced by the large number of measurements of longleaf pine and agricultural fuels which had high EF values.

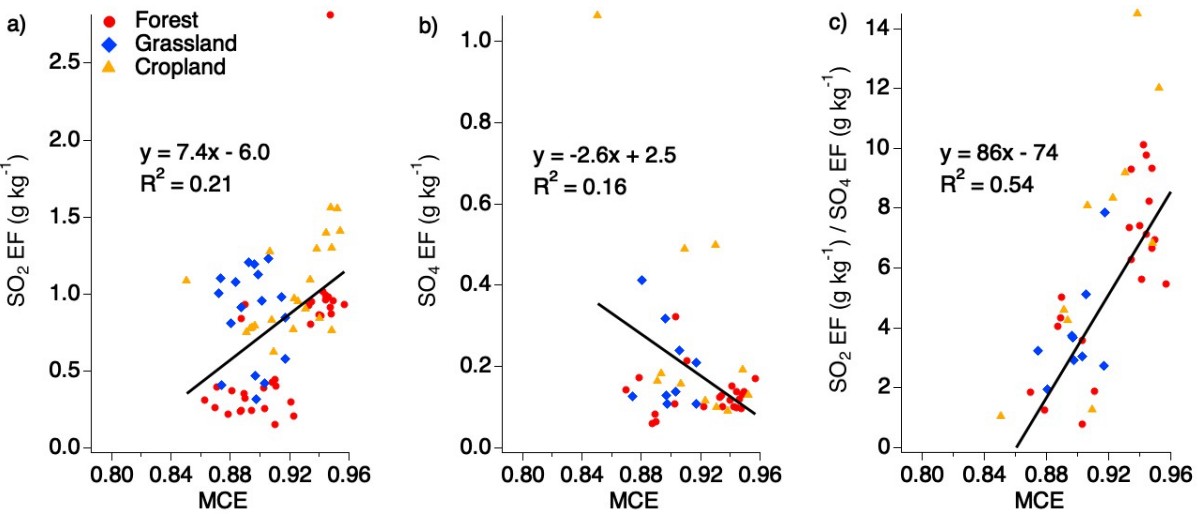

Figure 3. Scatter plots of EFs for $SO_2$ (a), sulfate (b), and the ratio of $SO_2$ to sulfate (c) (within 1 hr downwind of each fire source) vs. MCE based on combined fuel types.

Looking more closely at the different fuel types in comparison to the categories compiled by Andreae (2019), we see good agreement within the combined variability (Table 1 and Fig. 4). While the fuel types are categorized differently in this study, many still fit the characteristics of the categories listed in the compilation report allowing for comparison. Of the FIREX-AQ categories that allow for comparison with Andreae (2019), all EF data available are for the flaming stage.

The generally strong agreement between FIREX-AQ EFs and those in published inventories lends confidence to the quality of EFs underlying model emissions. Agricultural burns exhibit the highest EFs. This was reported by Andreae (2019) as $0.80 \pm 0.71$ g $kg^{-1}$ C in the flaming stage, similar to $1.1 \pm 0.30$ g $kg^{-1}$ C reported here. The temperate forest category, comprised here of forest and slash, produces a combined EF of $0.70 \pm 0.51$ g $kg^{-1}$ C which is in excellent agreement with the Andreae (2019) value of $0.7 \pm 0.48$ g $kg^{-1}$ C. Combining savanna, shrubland, grassland, and understory into the savanna/grassland category produces the largest difference in which the FIREX-AQ value of these combined fuels is $0.70 \pm 0.26$ g $kg^{-1}$ C, whereas, Andreae (2019) reported a value of $0.47 \pm 0.44$ g $kg^{-1}$ C; however, these values fit within the standard deviation.

Table 1. Comparison of the flaming stage $SO_2$ EFs (g $kg^{-1}$ C) by fuel type as measured during FIREX-AQ (left) to the compiled values reported in Andreae (2019) (right).

| Fuel Type (FIREX-AQ) | EF | StDev | Num tran[+] | Combined Categories | EF | StDev | Num stud[*] | Fuel Type (Andreae, 2019) |
|---|---|---|---|---|---|---|---|---|
| Forest | 0.66 | 0.49 | 35 | 0.70 ± 0.51 | 0.7 | 0.48 | 5 | Temperate forest |
| Slash | 1.15 | 0.38 | 3 | | | | | |
| Savanna | 0.47 | 0.06 | 2 | 0.70 ± 0.26 | 0.47 | 0.44 | 12 | Savanna/grassland |

| | | | | | | | | |
|---|---|---|---|---|---|---|---|---|
| Shrubland | 0.56 | | 1 | | | | | |
| Grassland | 0.83 | 0.29 | 6 | | | | | |
| Understory | 0.53 | | 1 | | | | | |
| Cropland | 1.09 | 0.30 | 16 | | - | 0.8 | 0.71 | 10 | Agriculture |

430    [+]Num tran indicates the number of transects measured within 1 hr downwind of the fire source measured during FIREX-AQ.
[*]Num stud indicates the number of studies included in the Andreae (2019) compilation.

The categories measured during FIREX-AQ that do not overlap with the Andreae (2019) compilation reflect smoldering conditions. For the most part, the majority of the smoldering stage $SO_2$ EFs exhibit lower values than the flaming stage by approximately 21—63% (Fig. 4). 435  The two FIREX-AQ categories (grassland and understory) which show smoldering $SO_2$ EFs to be larger than the flaming stage suggest the need for additional measurements to build statistical confidence.

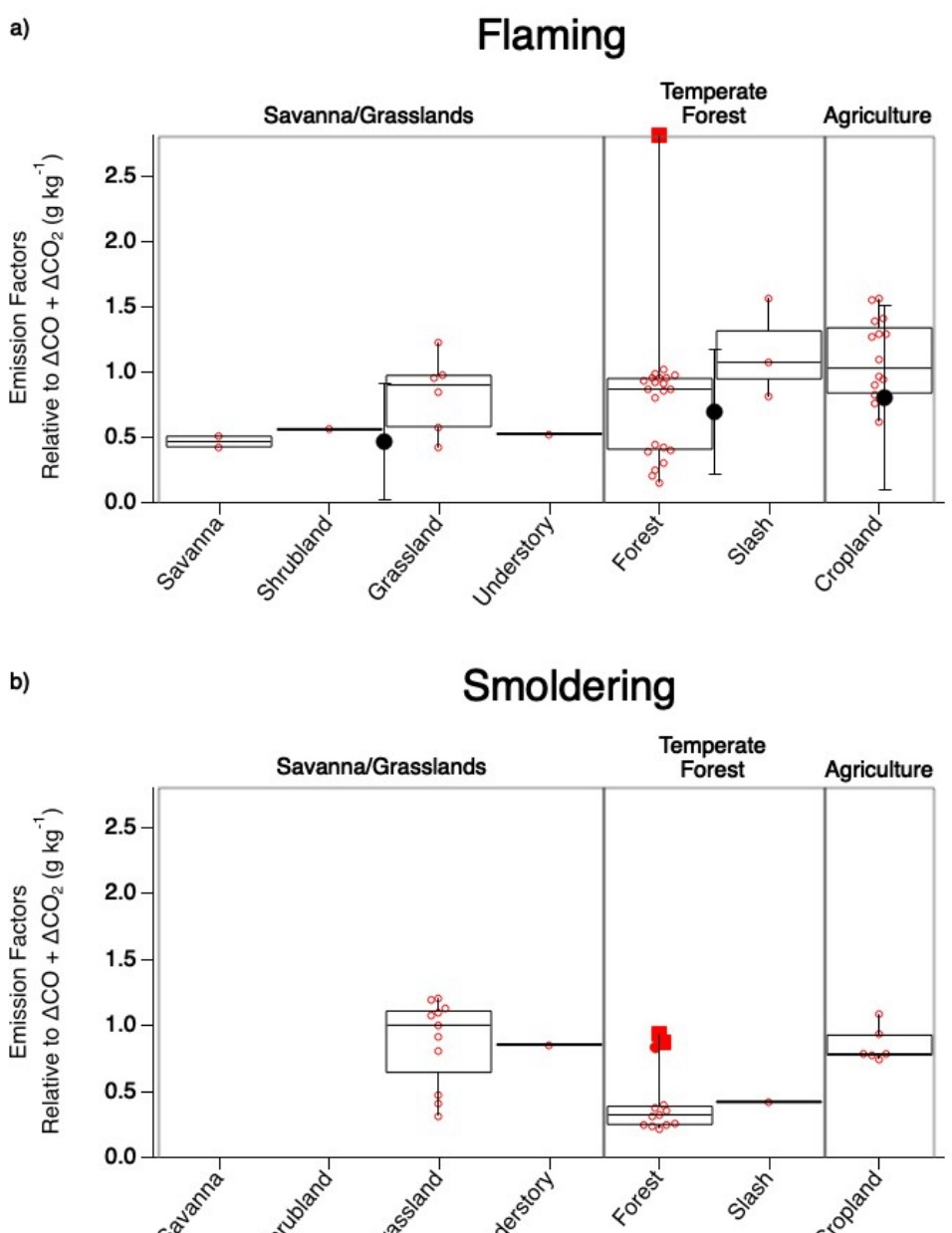

Figure 4. Comparison of $SO_2$ EF values observed during flaming (a) and smoldering (b) combustion across fuel types sampled during FIREX-AQ. The box upper edge represents the $75^{th}$ percentile and the lower edge the $25^{th}$ percentile with the median shown by the middle line. The whiskers represent the minimum and maximum observed values with the open circles representing each observation and the solid red circle representing a potential outlier. The large solid black circles with error bars depicting 1 standard deviation in panel (a) show corresponding average Andreae (2019) values.

### 3.2 Emission factor variability

The variability observed amongst the different fuel types may partly reflect variability in surface S content stemming from wet and dry deposition. Although this source of sulfur has significantly
decreased in the U.S. over the last two decades, the highest emission factors during FIREX-AQ were observed within the regions of the U.S. that typically experience the largest sulfur deposition rates as reported by the National Atmospheric Deposition Program (2022) (Fig. 5).

     Sulfur-containing fertilizers may also enhance S content in smoke. Sulfur aids plant uptake of nitrogen, and decreasing sulfur deposition over the last two decades has led to an
increased use of sulfur additives in fertilizers (Hinckley et al., 2020). Hinckley et al. (2020) report this sulfur application to range from around 20—300 kg S ha$^{-1}$ yr$^{-1}$, which occurs in the form of inorganic sulfate or elemental sulfur (Solberg et al., 2011). Given that the average yield of corn within the U.S. is 168 bushels per acre, a sulfur application of 20 kg S ha$^{-1}$ yr$^{-1}$ would result in 12 g S kg$^{-1}$ C in its composition. Assuming 10% of this added sulfur remains after
harvest and runoff and is present in the residual material that is burned, the remaining 1.2 g S kg$^{--1}$ could in part explain the enhanced emission factors in those regions (U.S. Department of Agriculture, 2020). Therefore, the observed variability in emission factors throughout the U.S. may be in part explained by the sulfur availability to the plants and soils, either from deposition or fertilizer use, resulting in larger emission factors from certain locations when burned.

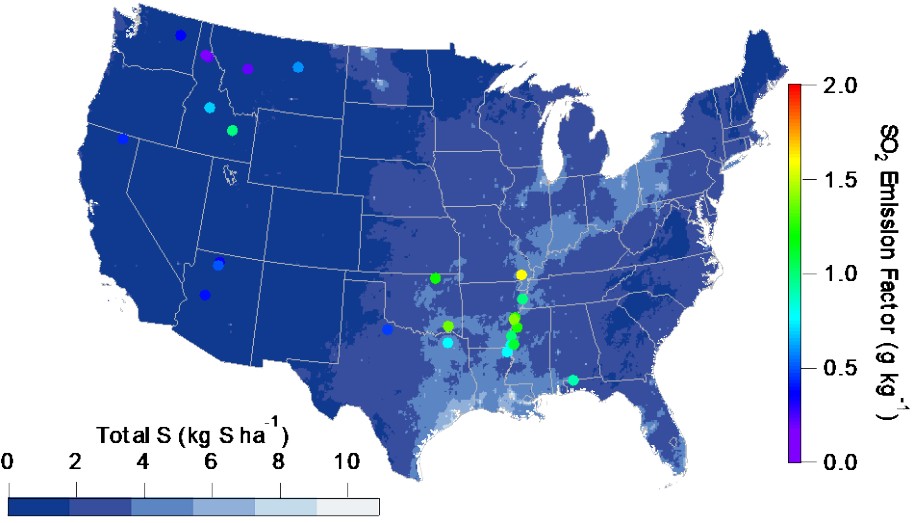


*Figure 5. National Atmospheric Deposition Program (2022) reported sulfur deposition rates (https://nadp.slh.wisc.edu/committees/tdep/#tdep-maps) compared to SO₂ EFs (closed circles) by geographical location as measured during FIREX-AQ for all fuel types.*

**4 Chemical Evolution of Sulfur**

After emission, SO₂ oxidizes to sulfate via both gas- and condensed-phase processes. Discrepancies reported by previous studies of modeled sulfate compared to measurements suggest that the conversion chemistry of SO₂ to sulfate is not fully understood. In this section, we
combine FIREX-AQ observations with a detailed chemical box model to evaluate the chemical mechanisms of SO₂ to sulfate conversion.

**4.1 Temperature dependence of sulfate production efficiency**

The balance of gas and particle phase sulfur between $SO_2$ and sulfate exhibits a marked temperature dependence amongst the cumulative flights while remaining generally constant during individual flights (Fig. 6). The fewer observations at temperatures below 265 K is the result of the range of aircraft altitude sampledsampled during this study. However, the decreasing trend shown by the numerous measurements between 265—283 K support the

suggestion of lower $SO_2$ concentrations compared to sulfate at the lower temperatures. Sulfate is >90% of the sum at temperatures below 265K, while above 285 K $SO_2$ and sulfate are equally balanced which is likely due to the quasi-second order process of heterogeneous oxidation in a plume (Freiberg, 1978). The noisy, but overall positive trend between 265—283 K suggests rapid chemistry after emission. Conversion of $SO_2$ to sulfate generally increases with decreasing

temperature due to increased aerosol water content and $SO_2$ and oxidant solubility, but the rapid change observed in this temperature regime also requires aqueous phase sulfur oxidation (Pattantyus et al., 2018).

         The majority of sulfur oxidation occurs in the aqueous phase. As observed during the 3 August flight, calculation of the contribution of OH to the decrease in $SO_2$ by applying an OH

concentration of $2 \times 10^6$ cm$^{-3}$ (Liao et al., 2021) produces a negligible $SO_2$ decay compared to the dilution normalized mixing ratio of $SO_2$ (Fig. S2). Similar behavior is expected for other flights due to similar conditions of limited photolysis near the center of the smoke plume.

         Recent studies have suggested HCHO to be an important aqueous phase oxidant at reduced temperatures (Moch et al., 2018; Song et al., 2021). However, HCHO is also an

indicator of smoke age with mixing ratios typically being largest nearest to the fire source (Liao et al., 2021). Considering measurements acquired when the HCHO mixing ratio is high (> 25 ppb), implicitly filtering out aged smoke, the slope of the $SO_2$ to total sulfate ratio over the 265––283 K temperature regime (0.04) shows a stronger correlation with temperature ($R^2$=0.74) (Fig. 6b, black line). Further limiting the effect of chemical aging by analyzing only those

measurements within 1 hr of the fire source, the conversion of $SO_2$ to sulfate is observed to be approximately 65% slower (Fig. 6b, red line) in the 265—283 K temperature range. This is consistent with heterogeneous chemistry in that aging occurs more rapidly at higher temperatures. While sulfate measurements within 1 hr of the fire source could be due to primary emission, this is expected to be a small fraction compared to $SO_2$ as shown in Fig. 2 and primary

emission would not exhibit the temperature dependence observed here.

         Other sulfate species contributee to sulfur conversion during this temperature regime. There were several periods identified during these flights in which organosulfur species were recognized to be a significant fraction of the AMS sulfate measurement. These measurements only occurred within the temperature range 270—285 K. When organosulfur was present in

plume transects within 1 hr downwind of the fire source, the $SO_2$ to total S ratio decreased with decreasing temperature 23% faster than in transects of fresh plumes when organosulfur was not present.

         These findings emphasize the importance of temperature in combination with smoke age and organosulfur production on the conversion of $SO_2$ to sulfate and is further investigated in

section 4.2.1.

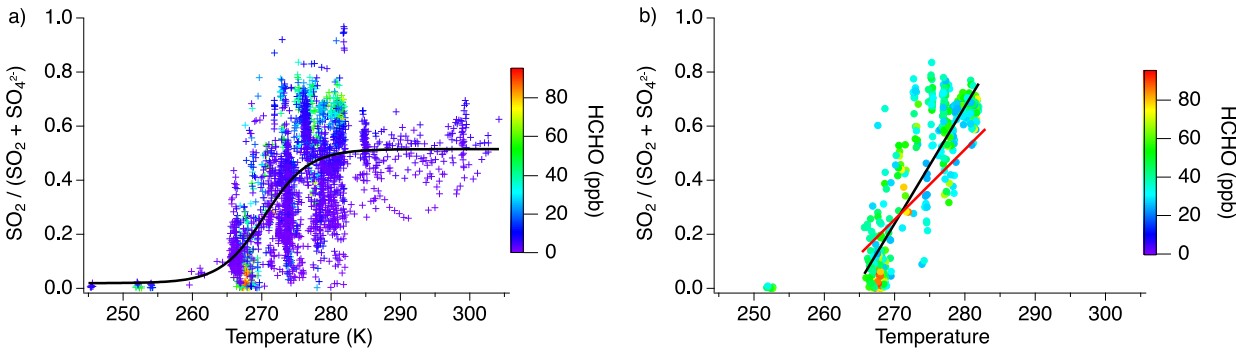

*Figure 6. Fractional sulfur conversion as a function of temperature a) including all smoke ages with a sigmoid fit and b) only measurements with HCHO > 25 ppb with the black line indicating the linear fit through the data at all ages between 265—283 K and the red line indicating the linear fit through the measurements within 1 hr of emission in the same temperature regime.*

## 4.2 Model results

### 4.2.1 Williams Flats 3 August 2019 flight

Select time series relating to the conversion of $SO_2$ to sulfate for the 3 August 2019 flight are shown in Fig. S3. Altitude and temperature were constant, around 3 km and 280 K, for both passes of about 10 transects each. Actinic fluxes trended downward for the second pass as dusk approached. Thermodynamic modeling suggests an average pH value of 5.3 (range of -2 to 8) over the length of the plume transects, but a possible increase in LWC by a factor of 2—3 during the second pass with an average of $2 \times 10^{-6}$ g sm$^{-3}$. Because the conditions of this flight are relatively consistent between passes, the measurements of both passes are combined for comparison to the model with pH and LWC held constant. Modeling results of this flight with the inclusion of all known gas- and aqueous-phase S(IV) pathways (Table S1) are shown in Fig. 7 with a conservatively assumed 30% uncertainty shown. This uncertainty range encompasses the uncertainties associated with the mechanism of aqueous phase uptake and chemical rate constants occurring at the specified LWC and pH.

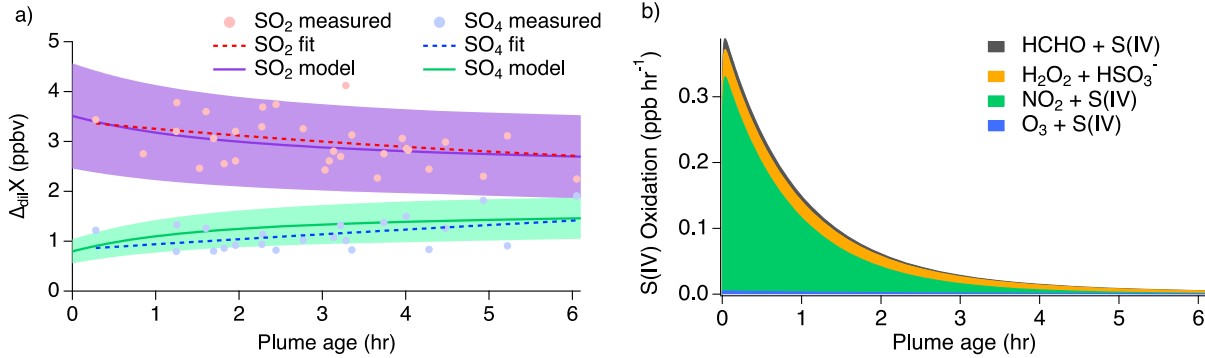

Figure 7. *(a) Dilution corrected ($\Delta_{dil}X$) measurementsm of 3 August 2019 shown by the markers and measurement fits shown by the dashed lines compared to the $SO_2$ and sulfate model results represented by the solid lines with shading denoting an estimated 30% model uncertainty. The*

*sulfate ($SO_4$) measurements represent total sulfate which potentially includes organosulfur. (b) Stacked modeled S(IV) oxidation rates leading to sulfate and HMS production.*

The model reproduces the general measurement trend of the 3 August flight for both $SO_2$ and sulfate (Fig. 7a). Model results for NO, $NO_2$, $NO/NO_2$, $O_3$, HCHO, and $H_2O_2$ are compared
to the measurements for each model in Fig. S4 showing good agreement for the 3 August flight. In accordance with the sulfate measurements, the modeled sulfate represents the sum of sulfate and HMS (the latter representing OS). A small, yet important, change is observed for the $SO_2$ and sulfate measurements with $SO_2$ decreasing by a linear slope of 0.15 ppb hr$^{-1}$ and sulfate increasing by a linear slope of 0.26 ppb hr$^{-1}$. The decrease in the S(IV) reactions (Fig. 7b) further
demonstrates this. The largest increase in these reactions is observed within the first 15 min, but the decrease in these reactions over the remaining 6 hrs indicates a slowing of this conversion. Under the conditions of this flight, the model indicates that aqueous phase oxidation by $NO_2$ and $H_2O_2$ are the dominant pathways leading to inorganic sulfate formation with little S(IV) reaction by HCHO and $O_3$ (Fig. 7b). This is in contrast to what has been previously expected of aerosol
S(IV) oxidation which has been thought to be dominated by ozone oxidation. However, the higher $NO_2$ oxidation rate constant with increased pH reported by Liu and Abbatt (2021) for non-ideal solutions increases the significance of this reaction.

**4.2.2 Williams Flats 7 August 2019 flight**

The 7 August 2019 flight shows distinct differences between the two passes (Fig. S5); therefore, the flight has been differentiated into the first pass (first full set of transects) and second pass (second full set of transects). It is also during this flight that the largest OS contribution has been reported for the AMS measurements during the FIREX-AQ wildfire flights.
The first pass was measured around 4 km and 276 K with an estimated dilution factor of approximately $8 \times 10^{-5}$ s$^{-1}$ and limited cloud presence. A pH of around 7.2 was estimated for this flight with an aerosol LWC of approximately $1 \times 10^{-7}$ g sm$^{-3}$. Both $NO_2$ and CO decrease at similar rates while HCHO remains relatively stable around 40 ppb and $O_3$ shows a decrease compared to the air outside of the plume for the first six transects (Fig. S6). $SO_2$ and sulfate are
fairly similar with a few instances of sulfate surpassing $SO_2$ in addition to a moderate fraction of OS observed during this pass.
The increasedincreased altitudeof the second pass is associated with an 8 K decrease in temperature relative to the first pass. The dilution factor for this pass was determined to be slower at around $3 \times 10^{-5}$ s$^{-1}$. The difference in these dilution factors could be due to measuring
at different altitudes or the result of a sampling artifact due to measuring in different sections of the plume, however, there is not enough information available to determine the exact cause. $NO_2$ appears to decrease more slowly in comparison to CO which remains relatively constant after the plume has moved away from the clouds. In addition, ozone, which shows the same trend as $O_x$, appears to be consumed more quickly in transects in which clouds were observed, suggesting
rapid uptake within the clouds, in addition to the fast reaction with NO producing the additional $NO_2$. This additional $NO_2$ in combination with limited photochemistry as a result of decreasing actinic flux (Fig. S7) due to approaching dusk conditions slows the decreasing $NO_2$ trend observed during this pass. Furthermore, ISORROPIA calculations indicate a 10-fold increase in aerosol LWC in the presence of clouds compared to the first pass. This is likely due to the
decrease in temperature (268 K) and larger relative humidity. The presence of clouds decreases

downwind concurrently with a decrease in relative humidity, but aerosol LWC remains high. Lastly, this pass shows SO₂ is nearly depleted in the center of the plume (Fig. S5) while sulfate increases substantially with a rather significant fraction of OS being observed (Fig. S8).

Due to these distinct differences between passes, each pass was modeled separately with the OS contribution reported independently from the sulfate measurements and model results. The modeled oxidation compounds (Fig. S4) show generally good agreement with the measurements for these passes; however, some discrepancies are observed due to measuring different parts of the plume. Results of the first pass are shown in Fig. 8 and the second pass shown in Fig. 9; both show good agreement between the model and measurements with ozone and NO₂ as the largest contributors to sulfate production during this flight. However, the majority of modeled S(IV)reaction occurs through the HCHO pathway rapidly producing HMS.

The first pass shows SO₂ increasing downwind, which is unexpected because SO₂ is considered to be a primary emission which typically decreases downwind as it is removed through oxidation. In addition, the measurements show a large OS mixing ratio following the first hour after emission before gradually decreasing downwind. This suggests that OS is either directly emitted from the fire source or very rapidly produced.

Clouds and large LWC were present throughout the majority of the second pass measurements (Figs. S5 and S6), significantly shifting the chemistry from that of the first pass. Figure 9 shows that modeled SO₂ is quickly taken up into the aqueous phase under higher LWC conditions $(6 \times 10^{-5}(6 \times 10^{-5}$ g sm$^{-3})$ and pH (7.2) with approximately 1.5 ppb going directly into sulfate production and the remaining 3 ppb of the initial SO₂ concentration being converted into HMS. Thesereaction processes occur promptly after emission, but they rapidly slow once all of the available initial SO₂ is depleted within the first 1—2 minutes. The exponential trends of the sulfate and OS measurements agree with the model results to within approximately 40%.

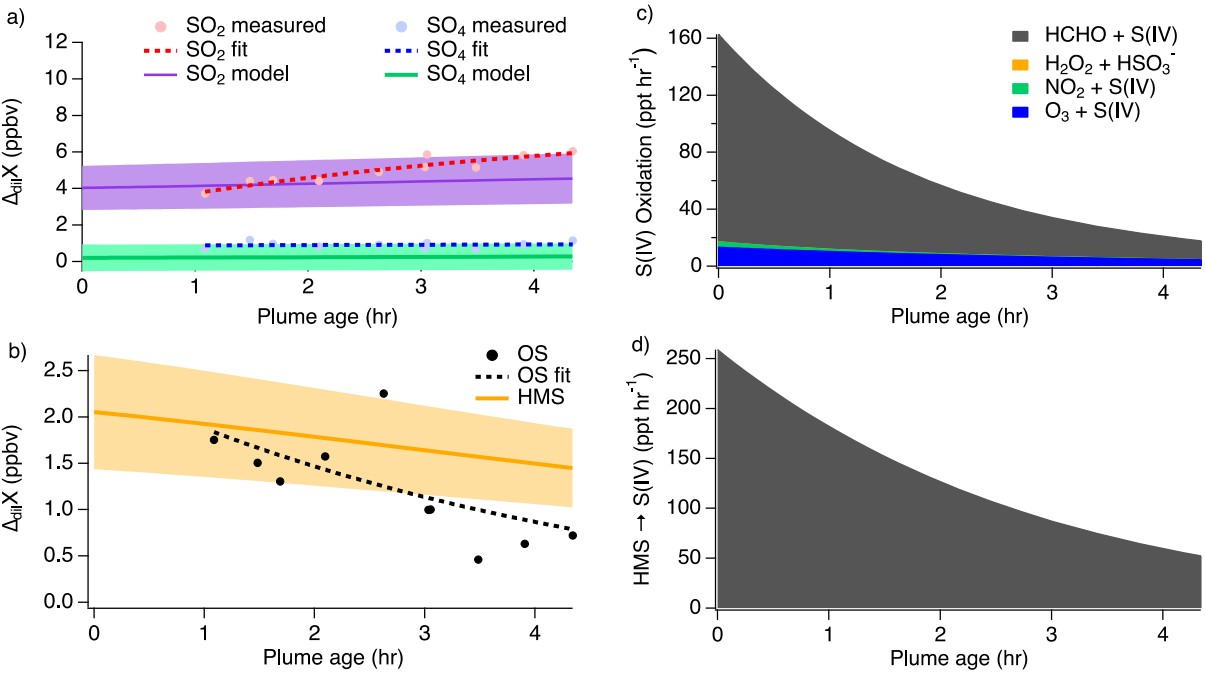

Figure 8. *First pass dilution corrected (Δ_dilX) measurements shown by the markers and measurement fits shown by the dashed lines compared to the model results represented by the*

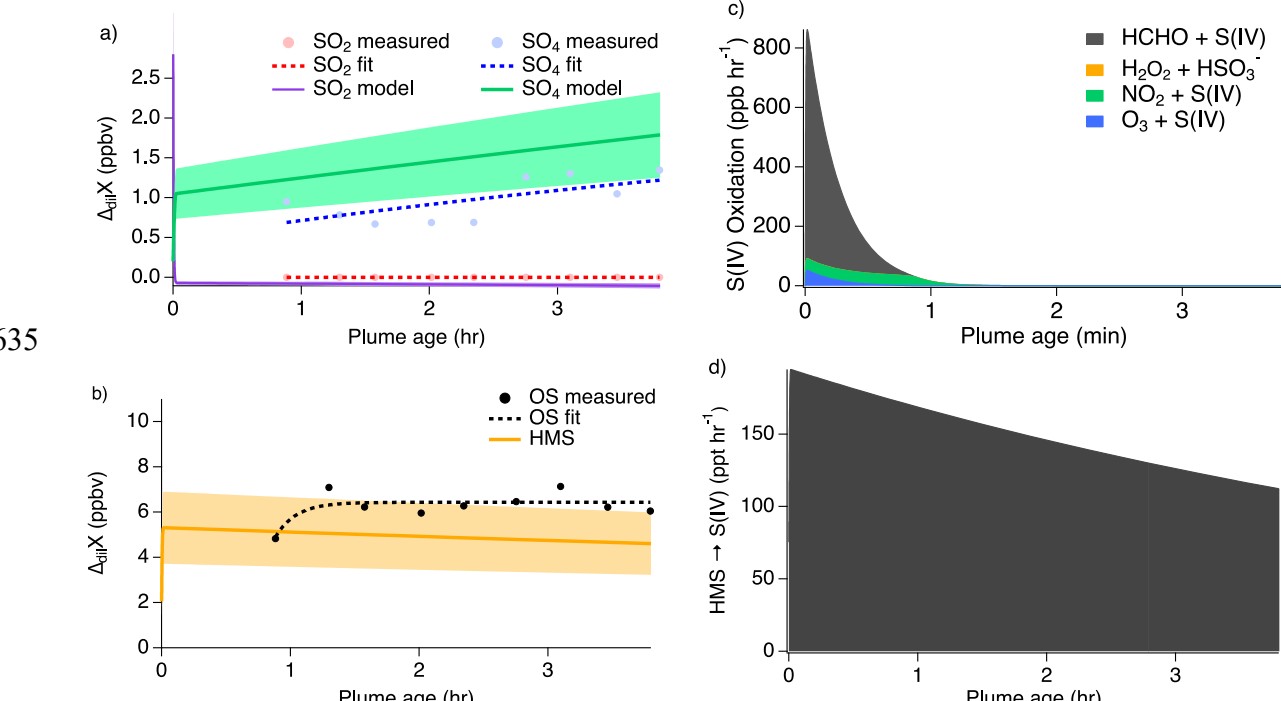


*Figure 9. Second pass dilution corrected ($\Delta_{dil}X$) measurements shown by the markers and measurement fits shown by the dashed lines compared to the model results represented by the solid lines with shading denoting an estimated 30% model uncertainty for SO₂ and sulfate (a)*

*and OS (b). Stacked modeled S(IV) oxidation rates (c) leading to sulfate and HMS production. HMS reverse reaction rate (d) reproducing S(IV).*

Comparing the 3 August and 7 August flights, the main differences leading to the different S(IV) reaction pathways are the pH and HCHO mixing ratios. Average pH on the 3

August flight was 5.3; whereas the 7 August flight experienced neutral conditions with a pH around 7.2. The initial HCHO mixing ratio was estimated to be 30 ppb for the 3 August flight and 50 ppb for the 7 August flight. While liquid water content plays a significant role in affecting the HMS reversal rate, each of these flights remained within the wet aerosol characterization with a calculated LWC of $2 \times 10^{-6} 10^{-6}$ g sm⁻³ for the 3 August flight and $66 \times 10^{-6} 10^{-6}$ g

sm⁻³ (4 km) and $66 \times 10^{-5} 10^{-5}$ g sm⁻³ (5 km) for the 7 August flight. The total S observed for these flights, in terms of SO₂ and sulfate show values of 2-10 ppb on average above the background; however, in the presence of organosulfates, this total S can increase to up to 15 ppb on average above the background.

The importance of HMS as a S(IV) reservoir and its conversion into sulfate or into gas-

phase SO₂ largely depends on the varying conditions of LWC. Under neutralized conditions (7.2), the model reproduces the observed trends of all three compounds under these wet aerosol conditions. As discussed further in section 4.2.3, the higher pH of this flight increases the rate of HMS reversal back into S(IV) by a factor of six. Because of the low LWC of the first pass, heterogeneous uptake is limited and causes the rates of S(IV) reaction to significantly decrease.

S(IV) evaporation then enhances gas phase $SO_2$ in transported smoke, consistent with similar rates of HMS decay and $SO_2$ growth. As a result, very little sulfate is produced during this pass at a rate of approximately 4 ppt hr$^{-1}$ primarily due to S(IV) oxidation by ozone. However, the higher LWC conditions of the second pass allow S to remain in the aqueous phase. The small increase in sulfate of approximately 500 ppt over the course of the flight can be explained by a

small fraction of HMS, on the order of 120—190 ppt hr$^{-1}$, which undergoes a reverse reaction decomposing back into S(IV) before being oxidized to produce sulfate (Fig. 9d).

        The SAGA-MC instrument detects HMS as S(IV), which cannot be separated from $HSO_3^-$ and $SO_3^{22-}$ and is therefore subject to an interference from high concentrations of gas-phase $SO_2$. However, the S(IV) from the SAGA-MC is comparable to the SAGA filter samples,

which are unaffected by ambient $SO_2$ and hence suggests that a large fraction of the S(IV) in the SAGA-MC was present in the aerosol, and that the contribution of the $SO_2$ artifact to the S(IV) signal is small. This observation further suggests that most of the S(IV) was present in submicron particles, as supermicron particles are not quantified by the SAGA-MC (Guo et al, 2021). As shown in Fig. S9, SAGA-MC sulfate measurements show similar concentrations to the AMS

inorganicinorganic sulfate measurements during bothboth passes. The AMS total sulfate is slightly larger than the SAGA-MC sulfate in the first pass, but considerably larger during the second pass. The SAGA-MC S(IV) (reported as $SO_3$) was similar to AMS $SO_{4,,org}$ on the first pass, but did not increase with AMS $SO_{4,,org}$ during the second pass suggestingsuggesting that HMS may have been the majority of the organosulfur concentrations measured during the first

pass but that an additional unknown organosulfur was much more abundant than HMS during the second pass. Therefore, it appears that the modeled HMS exceeds measurements on the second pass.

        There are two potential explanations for the good agreement between the observed organosulfur concentration from the second pass and the modeled HMS. It is possible that during

the very rapid uptake of $SO_2$ into the aqueous phase, (1) additional organosulfur species may be produced or (2) the additional organosulfur species are the result of further reactions of HMS suggesting that the model is correctly reproducing the HMS formation chemistry, but indicating that the model aqueous phase chemistry is incomplete. Both of these potential explanations require that the measured organosulfur species behave similarly to HMS in their rates of

formation and termination in order to explain the good agreement between the modeled HMS and measured organosulfur concentrations. In addition, these explanations would require that the organosulfur species are not identified as S(IV) in ion chromatography measurements. It is a potential possibility with the large mixing ratios of HCHO and $H_2O_2$ observed in these fire plumes that the chemistry of hydroxymethyl hydroperoxide as a result of HCHO and $H_2O_2$

reaction could be influencing the organosulfur production and should be considered in future studies (Dovrou et al., 2022). While the modeling allows for significant insight into the identity and formation mechanisms of aerosol sulfur, there is not enough evidence available from these measurements to conclusively explain all of the AMS and SAGA MC sulfur observations.

**4.2.3 Model HMS sensitivity analysis**

We performed a model sensitivity analysis to investigate the relevance of organosulfur behavior under the conditions of the HMS rates of production and termination in different environments by varying the model LWC ($10^{-6}$ – 1 g sm$^{-3}$), pH (1—8), temperature (260—280 K), and HCHO

(10—90 ppb) individually while holding the other parameters constant at the 3 August flight

conditions (T = 280 K, pH = 5.3, and LWC = $2 \times 10^{-4}$ g sm$^{-3}$) due to the more simplified chemistry occurring during this flight.

Variations in LWC (Fig. 10a) show that aerosols with less LWC produce minimal amounts of sulfate and HMS, but that HMS makes up between 5 and 45% of the combined concentrations. The HMS fraction shows the largest contribution as LWC increases into the cloud regime at which point sulfate production begins to decrease with a rapid increase in HMS. While the typical LWC range estimated for these fires is $10^{-7} - 10^{-2}$ g sm$^{-3}$, this indicates that the chemistry of the smoke will change substantially with cloud interactions. LWC is shown to be an important variable in the ratio of the formation of HMS to sulfate; however, this ratio trend is indicative of conditions at pH 5.3 and will vary under differing pH conditions.

The pH dependence of the ratio of HMS / (SO$_4$ + HMS) is shown in Fig. 10b in which HMS formation is more active as the acidity decreases. At acidic pH values, representative of typical tropospheric aerosol (Nault et al., 2021), a negligible amount of HMS contributes to the combined concentrations. Above pH 4, HMS contribution begins to increase followed by a more rapid increase after pH 6. The maximum HMS contribution is reached around pH 7.3 before rapidly decreasing at higher values.

The ratio of HMS production and reverse reactions varies with pH with the reverse reaction becoming more substantial at higher pH (Fig. 11). Under aerosol LWC conditions, the rate of the HMS reverse reaction is up to 3 times larger than the rate of HMS production. As LWC increases into the cloud regime, the rate of the HMS reverse reaction increases further to approximately two orders of magnitude larger than HMS production around pH 7. However, a reduction in temperature shifts this dependence to higher pH decreasing the rate of HMS reversal at the same pH.

While temperature and HCHO concentration are key factors controlling HMS production, these factors alone under low LWC and pH result in minimal HMS (Fig. S10S10). HMS production increases with decreasing temperature; however, under the conditions of the 3 August flight, HMS only reaches a maximum value of 5 ppt at 260 K which is approximately 5% of the modeled sulfate. Similarly, a minimal amount of HMS is produced with varied HCHO, but the ratio of HMS to the sum of HMS and sulfate increases linearly with HCHO at a rate of 1.5 ppt ppb$^{-1}$ HCHO.

The conditions that most largely affect HMS are LWC and pH. Due to the significance of LWC to HMS production and reversal, it is likely that aqueous aerosols, fog, cloud droplets, and possibly ice crystals will be most impactful on HMS production. Because the rainwater pH of areas such as the Western U.S. and Eastern China can reach much less acidic pH levels due to increased ammonia emissions, it is likely that these areas will be more susceptible to HMS production (Keresztesi et al., 2020; Qu and Han, 2021). Together, these conditions indicate that highly polluted areas which experience higher pH and greater LWC will likely be influenced by this chemistry. Therefore, the production of HMS should be an important consideration for air quality in areas such as agricultural regions which experience enhanced emissions of ammonia, likely increasing the pH, as well as geographical locations which may promote fog formation. This would include areas such as Beijing, the Uinta BasinBasin, and Bakersfield, CA, which have observed severe haze formation and have the potential to be affected by HMS.

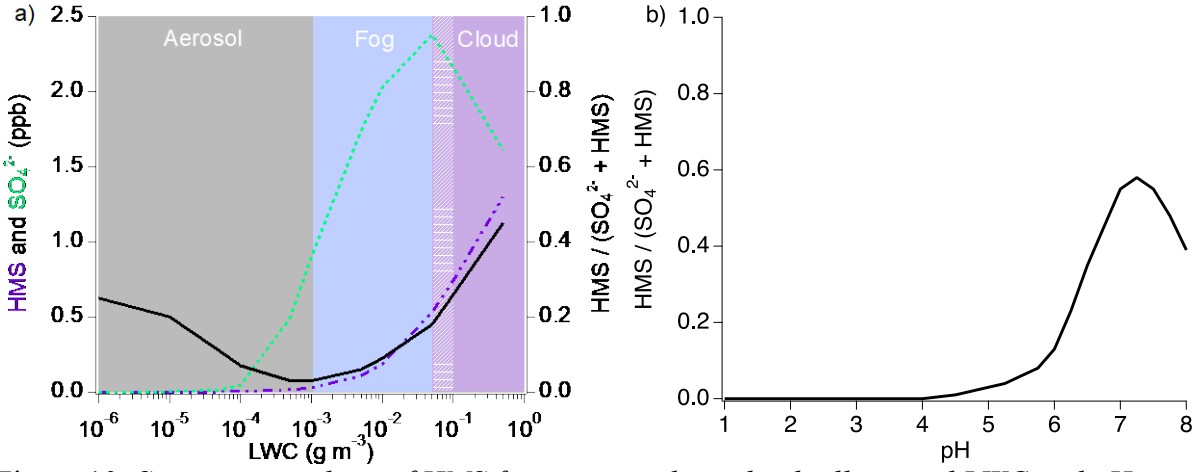

Figure 10. *Sensitivity analysis of HMS formation under individually varied LWC and pH conditions. The black line in each figure represents the ratio of the modeled HMS mixing ratio to the sum of the modeled inorganic sulfate and HMS. The shading in b) reflects the typical rainwater pH for each region.*

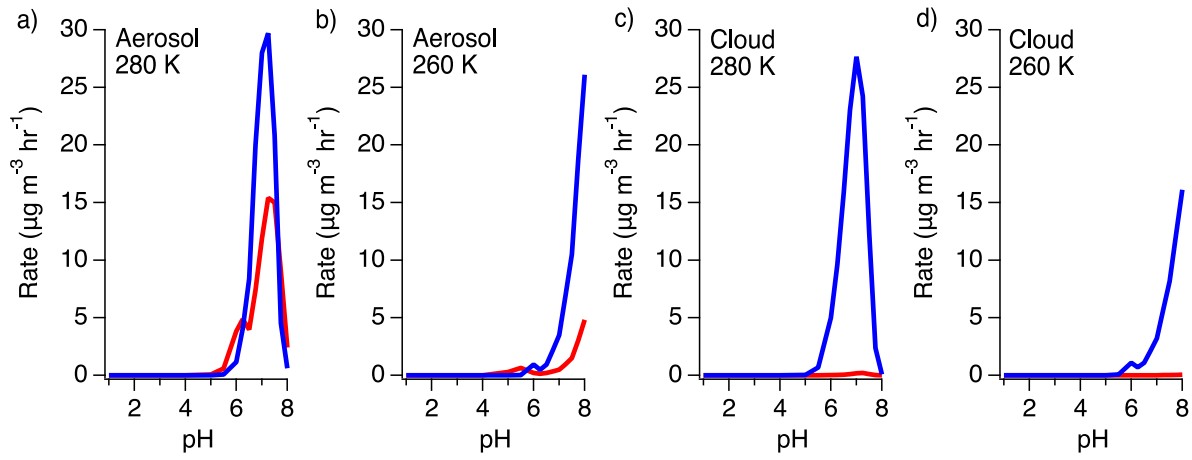

Figure 11. *Rates of HMS production (red) and reversal (blue) under aerosol and cloud conditions at 280 K and 260 K.*

## 5. Conclusions

SO₂ plays an important role in sulfate aerosol formation and thus air quality and climate forcing. Therefore, understanding the sources and evolution of SO₂ emissions in a changing climate are essential. The emission factors determined from the FIREX-AQ mission under flaming conditions show good agreement with the compilation reported by Andreae (2019). This provides confidence for the same categories under smoldering conditions for which there are no reported measurements from previous studies. No distinct correlation is observed for SO₂ emission factors based on MCE; however, it remains unclear if fire MCE influences the ratio of SO₂ and sulfate emission factors. With biomass burning events increasing worldwide, this study suggests that the resulting SO₂ emission factors will be more dependent on geographical location and land use, and less dependent on combustion phase and fuel type. Areas that incur more sulfur

deposition from coal burning or application through fertilizer use, will likely produce larger $SO_2$ emission factors.

Modeling with inclusion of the HCHOreaction chemistry, producing HMS, shows good agreement with the measurements. However, the differentiation of HMS from sulfate through the SAGA-MC measurements indicates that HMS can be over-predicted. While HMS is potentially directly emitted from the fire source, a large organosulfur concentration is observed that has not yet been identified. Because the modeled HMS is similar to the measured organosulfur fraction, it is expected that the additional organosulfur species likely exhibit similar rates of production
and termination as HMS. The importance of the HMS, or similar species, reverse reaction is also made apparent by the ability to act as an S(IV) reservoir. This allows these species to produce sulfate or $SO_2$ further downwind depending on the LWC and pH.

Environments that experience high LWC and pH are expected to be the most influenced by this chemistry. This includes regions that experience higher ammonia emissions and are
geographically or meteorologically subject to greater cloud or fog formation. As a result, this chemistry should be considered when assessing severe haze events as a result of either biomass burning or industrial pollution.

**Plain Language Summary**


Biomass burning sulfur dioxide ($SO_2$) emission factors range from 0.27–1.1 g kg$^{-1}$ C. Biomass burning $SO_2$ can quickly form sulfate and organosulfur, but these pathways are dependent on liquid water content and pH. Hydroxymethanesulfonate (HMS) appears to be directly emitted from some fire sources, but is not the sole contributor to the organosulfur signal. It is shown that
HMS and organosulfur chemistry may be an important S(IV) reservoir with the fate dependent on the surrounding conditions.

**Keywords**

Sulfur dioxide, hydroxymethanesulfonate, emission factors, biomass burning

*Data and code availability*. The data collected for FIREX-AQ are available from the NASA/NOAA FIREX-AQ data archive:  https://www-air.larc.nasa.gov/cgi-bin/ArcView/firexaq. The Framework for 0-D Atmospheric Modeling code is available from the AirChem/F0AM
archive:   https://github.com/AirChem/F0AM (doi.org/10.5281/zenodo.5752566).

*Author contribution*. The research was designed by PSR and AWR.  Measurement contributions were provided by all authors. The modeling was performed by PSR and GMW. The paper was written by PSR with contributions from all coauthors.

*Competing interests*. The authors declare that they have no conflict of interest.

*Acknowledgements*.  P.S.R. and A.W.R. acknowledge support from NASA's Upper Atmosphere Composition Observations program. MD, MS, and BW have received funding from the
European Research Council (ERC) under the European Union's Horizon 2020 research and innovation framework programme under grant agreement No. 640458 (A-LIFE), and from University of Vienna. HG, PCJ, and JLJ were supported by NASA 80NSSC18K0630 and

80NSSC21K1451 and NSF AGS-1822664. GMW, TFH, RAH, JMS, and JL acknowledge support from the NASA Tropospheric Composition program and the NOAA AC4 program (NA17OAR4310004). SRH and KU are funded under NASA grant 80NSSC18K0638. The National Center for Atmospheric Research is sponsored by the National Science Foundation. We would like to thank the NASA DC-8 crew and management team for support during FIREX-AQ integration and flights. Data from FIREX-AQ are available at (https://www-air.larc.nasa.gov/cgi-bin/ArcView/firexaq).

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
