# Peer review of "Emission factors and evolution of SO2 measured from biomass burning in wild and agricultural fires"

_Atmospheric Chemistry and Physics, 2022_

## Author Response (AR1)

*Response to the reviewers for the* "Emission factors and evolution of SO₂ measured from biomass burning in wild and agricultural fires" study *by* Pamela S. Rickly et al.

We thank the reviewers for their insightful comments which have helped to improve the manuscript.  Below are our replies to each comment in blue.

**Anonymous Referee #1**

**Comment #1:**  Lines 85 to 92, what is missing here is some idea on the relative contribution of biomass burning to overall sulfate in various regions. Eg, since this paper is about fires in the US, can references be sited or estimates made on the relative contribution of fires to total S near and long distances from the fires.

To the knowledge of the authors, there are no published observations of how total S biomass burning emissions contribute to overall sulfate for various regions.

**Comment #2:**  Line 160 and on: This measurement technique allows for the speciation of submicron non-refractory particulate mass and the direct separation of inorganic and organic species having the same nominal mass to charge ratio (DeCarlo et al., 2006; Canagaratna et al., 2007).   How is separation of inorganic and organic species possible if have same mass to charge?

A unit mass resolution spectrometer can determine the sum of ion signals at one particular m/z, making the inorganic signals interfere with organic signals or vice versa (e.g. NO+ and CH2O+ at m/z 30). As shown in the two quoted references, an AMS equipped with a high-resolution time-of-flight mass analyzer is capable of resolving these details  (e.g. see Fry et al 2018 for a detailed discussion of the practical implications for the nitrate and organic ions mentioned in the example above).

**Comment #3:**  Line 174 and on. What is the SAGA instrument (from Fig S8 it is the MC), which should be specified in the main text.  Also state if both instruments sample over similar particle size ranges or not, and if not, what is the possible effect?  What about comparing sulfate from the SAGA MC and Filters to see if there is substantial sulfate at higher particle sizes compared to those measured by the AMS and MC.

The SAGA instrument definition has been added at line 173.

The size coverage of different instruments is important in intercomparison and has been discussed in detail for these two instruments in Guo et al. (2021) for the ATom aircraft mission on the same airborne platform. Both AMS and SAGA MC/IC sample nominally submicron particles onboard the DC8 through different inlets and agree with each other in general, as shown in multiple comparisons over the years (e.g., Fisher et al. 2011). Despite some nuances between the two, (1) size selection of particles at ambient conditions (SAGA MC/IC) vs. dry (AMS); (2) the cutoff size is on aerodynamic diameter (SAGA MC/IC) vs. vacuum aerodynamic diameter

(AMS; due to the aerodynamic lens inlet of the AMS being the main source of large particle loss), (the following has been added at line 175) both MC/IC and AMS sample submicron particles with very similar effective size cutoffs for the range of altitudes sampled in the FIREX-AQ (van Donkelaar et al., 2008; Guo et al., 2021).

Supermicron sulfate is rare throughout FIREX-AQ, such as the Aug 3$^{rd}$ flight. However, substantial coarse mode sulfate, especially in the 2$^{nd}$ pass during the Aug 7$^{th}$ flight, was observed and up to several times higher than the submicron MC/IC sulfate.

**Comment #4:** Regarding uptake of gases, such as SO2, HCHO …. I assume (3.4) & (3.6) is uptake to the dry particles since the LAS reported dry distributions? If so, it is not clear how uptake is to ambient wet particles and equilibrium between gas-LWC is handled. For species that may react slowly in the particle or LWC phase, is equilibrium established between the gas and particles by this model? What are the time scales for equilibrium for the various species and what are the time scales for which concentrations are changing in the plume being modeled? Same applies for calculation of pH where an equilibrium model is used (ISORROPIA). In essence, is equilibrium assumed, and if so, is it that reasonable (ie, provide justifications).

The following clarification has been added at line 279:
The dry particle size (not ambient particle size) is incorporated into khet through Eq. 3.4. This khet value is then applied to Eq. 3.7 as a ratio to the LWC and ability of uptake (H), allowing for calculation of the gas-particle equilibrium. Therefore, as the particle size increases, greater condensation is able to occur, but this also allows for increased evaporation. However, with an increase in LWC and H, less evaporation will be expected. Using this method of uptake and evaporation does not allow for equilibrium of all processes to be assumed as it does in the ISORROPIA calculations.

ISORROPIA calculates pH and LWC at equilibrium. The required time scales to reach equilibrium after emission depends on the mass transfer rate, which is related to meteorological conditions, particle size, and phase state. For fine particles, the characteristic time for water equilibrium is very short (< 1 s) (Pilinis et al., 1989) and ranges from 20 mins or less (Dassios and Pandis, 1999; Cruz et al., 2000; Fountoukis et al., 2009; Guo et al., 2018) up to 10 hrs for semivolatile components, NH3, HNO3, and HCl (Meng and Seinfeld, 1996; Fridlind and Jacobson, 2000; Shingler, et al., 2016) (a slightly revised version of the sentence has been added at line 306). In the absence of highly viscous fine particles, the equilibrium assumption often holds in the ambient condition if not too close to the source. This also applies to the fire-emitted particles discussed in this study for aging over several hours assuming OA does not participate and is externally mixed. The agreement between predicted and measured gas-particle partitioning further validates this assumption.

**Comment #5:** Regarding the pH prediction (lines 290 and on), it would be useful to show the particle phase ammonium predicted and measured (and same for gas phase ammonia), this would give both an idea of the model prediction and the amount of data thrown out due to the 40% criteria. Cations are mentioned, but not in much detail, specifically, what about K+, which could be high and seems odd not to be in the pH calculation.

The following has been added to line 306:
The gas-particle partitioning is reproduced with ISORROPIA-II, with the regression slopes of predicted $NH_3$, $NH_4$, and $NO_3$ close to one compared to the observations and highly correlated (slopes: 0.949, 1.116, and 1.002; $r^2$: 0.991, 0.96, and 0.99996, respectively).

[Figure]

The 40% criteria throw out 4.6% of the data to improve data quality since these predicted particle pH are more uncertain. We have revised the modeled particle pH to that based on inorganic sulfate (vs. AMS total sulfate) and potassium (vs. zero non-volatile cations). The updated pH is on average 0.96±0.95 higher.

**Comment #6:** Lines 347 to 362: Maybe the lack of correlation with MCE is that the MCE dynamic range is small? One might try looking at BC/OA ratios, just out of curiosity.

Because the range of MCE values presented in Fig. 3 covers a similar range as used in the cited analyses, we believe this range is adequate for the determination of a lack of correlation.

**Comment #7:** Line 465 and Fig 6a. There are very few data points for T<265K. Please comment on why this is (eg, what is the aircraft altitude) and how this limits conclusions drawn from this plot. One may wish to make a plot of altitude vs SO2/total S, since alt and T are related. Could this be explained by instrument sampling artifacts as a function of altitude? If this trend is driven by liquid reactions, then test with looking at predicted liquid water vs T or altitude.

The following has been added at line 474:
The fewer observations at temperatures below 265 K is the result of aircraft altitude in which the lowest temperatures are correlated with the highest altitude of 6 km. However, the decreasing trend shown by the numerous measurements between 265-283 K support the suggestion of lower SO2 concentrations compared to sulfate at the lower temperatures.

**Comment #8:** Define vertical axis in Fig 7a fig caption.

This is now defined in the captions for Figs 7, 8, and 9 as dilution corrected ($\Delta_{dil}X$).

**Comment #9:** Line 521, what is the uncertainty here, if uncertainty in pH and LWC are considered? Eg, 30% is stated for the various measurements and chemistry (I assume) for a given pH and LWC, but how sensitive is this to pH and LWC? (This is a common issue with the whole modeling section). The average pH for this smoke plume was 5.3 (range -2 to 8), but the LWC data is not given (at least I did not see it), please provide. It would be interesting to note

the ratio of LWC to dry aerosol mass, which could be estimated based on the AMS data. This would give a sense of how wet the particles were.

The following has been added to Line 306 to address the uncertainty in pH and LWC:
The uncertainty in particle pH is estimated to be within 0.5-1 unit based on the sensitivity of pH to $NH_3$-$NH_4$ partitioning and and varies from point to point depending on the model reproduction of the partitioning (Guo et al., 2017). Because these calculations are based on the inorganic aerosol concentrations, additional uncertainties may arise from the organic phase (both from LWC and organic acids). The LWC could potentially be up to several times greater due to the dominant organic portion in the fire plumes despite the lower hygroscopicity compared to the inorganics (Kreidenweis et al., 2008; Guo et al, 2015; Brock et al, 2016). The mixing state of inorganic and organic for the particles in the early phase plumes remains to be investigated but is likely to be phase separated given the low oxidation state of the organics (Sullivan et al., 2020). The current modeling can be interpreted as assuming a phase separation of inorganic vs. organics, with the chemistry studied occurring only in the inorganic-dominated phase and its associated water, with no kinetic limitations due to a potential core/shell or micelle like structures present in the particles… Propagating the uncertainties of AMS inorganics (34%) and DC-8 aircraft RH sensor (3% based on the observed RH) gives an LWC uncertainty of 39% (Guo et al., 2015).

The LWC used for this flight was ($2 \times 10^{-6}$ g m$^{-3}$). This has been added to the discussion of the results. The dry aerosol mass for each transect of the 3 August flight was measured to be 5-50 µg sm$^{-3}$. As a result, the ratio of LWC to dry aerosol mass was 4-40.

The uncertainty of the model results is discussed in response to comment #12.

**Comment #10:** Why not make plots of pH and LWC vs plume age for all these modeled plumes, given that these are key variables?

We believe that the trends of these variables are clear in the data presented in the supplementary materials.

**Comment #11:** Also in Fig 7, what is the relative change in SO2 and sulfate in the 6 hrs? It appears to be fairly small. Fit the SO2 and sulfate data vs time with a line and test if the slope is actually not zero. Fig 7b, kind of gives a false sense of the importance of SO2 oxidation since it shows a huge time dependance, but in reality, there is very little change.  It would be more realistic to plot the vertical axis as rate of oxidation divided by total sulfate, or something similar.

The following has been added at line 551:
A small, yet important, change is observed for the SO2 and sulfate measurements with SO2 decreasing by a linear slope of 0.15 ppb hr-1 and sulfate increasing by a linear slope of 0.26 ppb hr-1. The decrease in the S(IV) reactions (Fig. 7b) further demonstrates this. The largest increase in these reactions is observed within the first 15 min, but the decrease in these reactions over the remaining 6 hrs indicates a slowing of this conversion.

**Comment #12:** Regarding Section 4.2.2. Similar questions apply as noted above regarding sensitively of predictions to pH and LWC. What species is driving the pH so high (7.2)? Is this a realistic particle pH? The LWC is 10^-7 g/m3; what fraction of dry particle mass is that? Say the particle mass is 100 ug/m3, that is a ratio of 0.1, a small amount of LWC.

The uncertainty in the modeled SO2 and sulfate due to pH is difficult to calculate due to the S(IV) mole fraction pH dependence. Similarly, LWC increases the solubility of all relevant compounds, but the conversion of SO2 to sulfate will rely on the concentrations of the available oxidants that enter the aqueous phase. As for HMS, the uncertainty due to changes in these parameters can influence the rates of production and reversal of HMS as reported in Fig. 11. Therefore, only the uncertainty in the measurements and chemical reactions is reported in the figures describing the chemical evolution downwind of the fire source.

This is one of the highest submicron pH for ambient particles to the authors' knowledge, even higher than the winter haze conditions in China. Several factors drive the particle pH in the fire plumes so high: (1) very high NH3 (on average $1.7 – 26$ µg sm$^{-3}$ in the plumes); (2) high NO3 and almost no HNO3; (3) the presence of non-volatile cations. The NH3 was several times higher than that of some winter haze events in China (Wang et al., 2016; Liu et al., 2017), and a ten-fold increase in NH3 makes pH one unit higher (Guo et al., 2017). Furthermore, the NH3-NH4 partitioning was mostly dominated by the particle phase, which corresponds to a higher pH generally speaking (Guo et al., 2017). The pH predictions were validated by reproducing the gas-particle partitioning as discussed in the thread of Comment #5 above.

**Comment #13:** Line 595, agree within 40% for what pH and LWC?

This has been clarified on line 610 as a LWC of $6 \times 10^{-5}$ g m$^{-3}$ and pH 7.2.

**Comment #14:** In the second pass, how did the pH change relative to the first pass? Was it even higher due to higher LWC (x10)?

The pH was initialized in the model at 7.2 for each pass during the 7 August flight. This has been clarified on line 610.

**Comment #15:** The sensitivity analysis is unclear (section 4.2.3.). Figure 11 shows that no HMS will be produced since loss is much larger than production, but the conditions of this flight (Aug 3) showed little organo-sulfate, so I guess there is consistency. Since highest concentrations of organo-sulfate was observed on the second leg of the 7 Aug flight, why not do a sensitivity analysis for this data. In fact, I would focus on this data, showing the predicted pH and LWC in detail and possibly also show how well ISORROPIA does by providing a comparison of the partitioning of ammonia, ammonium, nitric acid, and nitrate for this data. This could provide much more insight than the current analysis.

Because of the better agreement between the model and observations for 3 August, this flight was chosen for this sensitivity analysis. With the varied conditions of pH, LWC, and temperature used, the authors believe this provides the appropriate overlap with the conditions of the 7 August flight.

Per the response to Comment #5, the ISORROPIA metrics have been added to the paper.

**Anonymous Referee #2**

**Comment #1:** The authors provide an HMS loss rate constant of: $6.2 \times 10^8 \times \exp(-11400 \times (1/T - 1/298)) + 4.8 \times 103 \times (Kw/ H+) \times \exp(-4700 \times (1/T - 1/298)$, while Song et al. (2021, ACP), which is used as the reference of the rate, provide a rate constant of: $6.2 \times 10^{-8} \times \exp(-11400 \times (1/T - 1/298)) + 4.8 \times 103 \times (Kw/ H+) \times \exp(-4700 \times (1/T - 1/298)$. I assume that this is a typo, but please clarify.

This is a typo in Table S1 and the reaction is correctly represented in the model as $k = 6.2 \times 10 -8$. The supplement has been corrected accordingly.

**Comment #2:** The role of HMS as S(IV) reservoir is very interesting, especially since this result is mainly under conditions of pH>6 and high LWC, in which HMS has been shown to be unstable and prone to additional reactions. In the model, the formation and decomposition of HMS is included (Table S1), however its reactions with OH and H2O2, which has been shown to occur at pH>6 (Kok et al., 1986. J. Geoph. Res.; Martin et al., 1989, Atmos. Environ.; Chapman et al., 1990, Atmos. Environ.) are not included. How are the results affected upon inclusion of these reactions?

The following has been added to the supplement at line 54:
While HMS is resistant to reaction with $H_2O_2$, aqueous phase reaction of HMS with OH radicals is likely to occur (Olson and Fessenden, 1992). However, aqueous OH is short-lived and when included in the model, through condensation and evaporation (Eqs. 3.6 and 3.7 in main paper), produced no change in the model results.

**Comment #3:** The model represents efficiently the field data of August 3rd, however it does not capture the trend of all the field data for the case of August 7th. Since both days are within the same campaign in Boise, ID, what was the main differences between these days? It would be interesting to provide a brief explanation on why the two days differ, as provided for the two passes of the 7th of August.

The following paragraph has been added at line 635:
Comparing the 3 August and 7 August flights, the main differences leading to the different S(IV) reaction pathways are the pH and HCHO mixing ratios. Average pH on the 3 August flight was 5.3; whereas the 7 August flight experienced neutral conditions with a pH around 7.2. The initial HCHO mixing ratio was estimated to be 30 ppb for the 3 August flight and 50 ppb for the 7 August flight. While liquid water content plays a significant role in affecting the HMS reversal rate, each of these flights remained within the wet aerosol characterization with a calculated LWC of $2 \times 10^{-6}$ g sm-3 for the 3 August flight which also occurs between the LWC for and the 7 August flight of $6 \times 10^{-6}$ g sm-3 (4 km) and $6 \times 10^{-5}$ g sm-3 (5 km) for the 7 August flight.

**Comment #4:** Field data are provided for mainly August 3rd and 7th, which correspond to the Boise flights. It would be beneficial to provide field data and the model performance for the Saline flights. Are the main results the same for both flights? This is not very clear.

Because the focus of the Salina flights was agricultural burns which had smaller plumes, only 1-2 transects were flown for each fire. As a result, there were not enough measurements of each agricultural burn measured during the Salina based measurements to extrapolate back to the fire source and perform modeling for those plumes as well.

**Comment #5:** It is stated in the manuscript that HMS can be over-predicted and that additional organosulfur species can be "the result of further reactions of HMS suggesting that the model is correctly reproducing the HMS formation chemistry, but indicating that the model aqueous phase chemistry is incomplete" (lines 639-641). The inclusion of HMS oxidation via OH and H2O2 might improve the HMS prediction for the cases that pH>6, however for more acidic conditions there is another pathway that can lead to sulfate formation but also add to the HCHO loading and potentially affect the HMS chemistry. HCHO can react directly with H2O2 forming hydroxymethyl hydroperoxide, which can then decompose to reform HCHO and H2O2 (Dovrou et al., 2022, PNAS). Since H2O2 and HCHO are observed via the flight measurements, could this pathway be useful for the model representation of these species as well as the organosulfur chemistry (as it provides further information regarding HCHO (source of HMS))?

The following has been added at line 676:
It is a potential possibility with the large mixing ratios of HCHO and $H_2O_2$ observed in these fire plumes, that the chemistry of hydroxymethyl hydroperoxide as a result of HCHO and H2O2 reaction could be influencing the organosulfur production and should be considered in future studies (Dovrou et al., 2022).

---

## Author Response (AR2)

We thank the reviewers for their further comments which help to clarify important details within the manuscript. Below are our responses to each follow-up comment in blue.

Follow-up comment #1

Initial Comment #1: Lines 85 to 92, what is missing here is some idea on the relative contribution of biomass burning to overall sulfate in various regions. Eg, since this paper is about fires in the US, can references be sited or estimates made on the relative contribution of fires to total S near and long distances from the fires.

Reply to Comment: To the knowledge of the authors, there are no published observations of how total S biomass burning emissions contribute to overall sulfate for various regions.

New Comment: Eg, even though SO2 emissions are discussed, the authors claim in the first line of the Abstract that; Fires emit sufficient sulfur to affect local and regional air quality and climate. I am simply asking to assess this. Since a large fraction of the SO2 goes to sulfate aerosol, and the sulfate is measure and discussed, why can't the sulfate (could include organo-sulfate) in plume be compared to out of plume (eg, data away from the burning regions) for data from this study. A simple statistical result could be given. Or look at in-plume versus studies that report sulfate throughout the US. Please seriously consider the comment. (eg, Hand, J. L., B. A. Schichtel, W. C. Malm, and M. L. Pitchford (2012), Particulate sulfate ion concentration and SO2 emission trends in the United States from the early 1990s through 2010, Atmos. Chem. Phys. , 12, 10353-10365.)

The following has been added at line 671:

The total S observed for these flights, in terms of $SO_2$ and sulfate show values of 2-10 ppb on average above the background; however, in the presence of organosulfates, this total S can increase to up to 15 ppb on average above the background.

Follow-up comment #2

Initial Comment #3: Line 174 and on. What is the SAGA instrument (from Fig S8 it is the MC), which should be specified in the main text. Also state if both instruments sample over similar particle size ranges or not, and if not, what is the possible effect? What about comparing sulfate from the SAGA MC and Filters to see if there is substantial sulfate at higher particle sizes compared to those measured by the AMS and MC.

Reply to Comment: The SAGA instrument definition has been added at line 173. The size coverage of different instruments is important in intercomparison and has been discussed in detail for these two instruments in Guo et al. (2021) for the ATom aircraft mission on the same airborne platform. Both AMS and SAGA MC/IC sample nominally submicron particles onboard the DC8 through different inlets and agree with each other in general, as shown in multiple comparisons over the years (e.g., Fisher et al. 2011). Despite some nuances between the two, (1) size selection of particles at ambient conditions (SAGA MC/IC) vs. dry (AMS); (2) the cutoff size is on aerodynamic diameter (SAGA MC/IC) vs. vacuum aerodynamic diameter (AMS; due to the aerodynamic lens inlet of the AMS being the main source of large particle loss), (the

following has been added at line 175) both MC/IC and AMS sample submicron particles with very similar effective size cutoffs for the range of altitudes sampled in the FIREXAQ (van Donkelaar et al., 2008; Guo et al., 2021). Supermicron sulfate is rare throughout FIREX-AQ, such as the Aug 3rd flight. However, substantial coarse mode sulfate, especially in the 2nd pass during the Aug 7th flight, was observed and up to several times higher than the submicron MC/IC sulfate.

New Comment: This needs more clarity since very little information is given on what the SAGA MC is reporting. From the wording I assume this is a sulfate, but does the MC-IC analysis also measure S(IV)? I assume the S(IV) is from dissolved SO2, but does SAGA MC collected SO2 in biomass burning plumes affect the measured sulfate? This would be in addition to conversion of particulate organo-sulfates being converted to inorganic sulfate as discussed. It seems it is not a big issue given the good comparison shown in Fig. S8.

What about the SAGA filter measurements of sulfate – the replay above does not address the question? Why are the SAGA filters not used to assess HMS (or S(IV) in general) vs sulfate? Is the data not available? Much of this manuscript is on a model predicting HMS, HMS data may exist, or it may not, and a number of readers of this paper will know that. It is thus strange that this is never mentioned. Adding a sentence or two stating (if it is true) that the SAGA filter data cannot be used to determine HMS would provide clarity (and the reason why or why not). The same applies to the AMS, see (Dovrou,et al, Atmos. Meas. Tech. , 12, 5303-5315). This paper is cited, but it is not explicitly noted why the AMS is not used to directly quantify HMS.

The SAGA filters have similar concentrations and trends of S(IV) as the MC, but they were not frozen after collection and thus we prefer to use the MC measurements as the primary ones.

We have modified the paragraph in the manuscript where the measurements are described to clarify these points including the method description of SAGA filter as follows:

Sulfate measurements were performed by a suite of in-situ instruments: an Aerodyne high-resolution time-of-flight aerosol mass spectrometer (AMS) (DeCarlo et al., 2006; Canagaratna et al., 2007), with a sampling rate of 1-5 Hz, the online soluble acidic gases and aerosol mist chamber (SAGA-MC) coupled with ion chromotograph (IC) (Scheuer et al, 2003; Dibb et al, 2003), with a sampling interval of 75 s, and SAGA filter collector with subsequent offline IC analysis (Dibb et al., 1999; Dibb et al., 2000). Both SAGA MC/IC and AMS sample submicron particles, while the SAGA filter collects both submicron and supermicron particles up to 4.1 µm with 50% transmission (McNaughton et al., 2007; van Donkelaar et al., 2008; Guo et al., 2021). The AMS instrument allows for the speciation of submicron non-refractory particulate mass and the direct separation of inorganic and organic species having the same nominal mass to charge ratio (DeCarlo et al., 2006; Canagaratna et al., 2007).  Both inorganic and organic sulfate fragment similarly in the AMS, mostly to $H_xSO_y^+$ ions without carbon. There can be some differences in fragmentation between organic and inorganic sulfur that can in some cases be used to separate organic from inorganic sulfate (Farmer et al., 2010). For AMS total nitrate, where the fragmentation pattern is similar (Farmer et al., 2010), techniques for rapid assignment of organic nitrate based on its fragmentation pattern have been successfully developed (Fry et al., 2013; Day et al., 2021).

 While there are some differences in fragmentation between organic and inorganic sulfur that have been used in some cases to separate organic from inorganic sulfate (Chen et al., 2019; Dovrou et al., 2019); the sulfate fragmentation pattern is overall much more variable compared to nitrate and hence such approaches will work only in very specific instances (Schueneman et al., 2021). In this work, we found the ion fragmentation method to produce reasonable results, based on the consistency with the results using positive matrix factorization (PMF, Paatero et al., 1994, Ulbrich et al., 2009) and the measurements of submicron sulfate aerosol  from SAGA-MC, which quantifies only inorganic sulfate.  The correlation between the AMS inorganic sulfate and SAGA-MC sulfate shows an overall good agreement (Fig. S8), which adds confidence to the AMS apportionment. However, as discussed in section 4.2.2, for certain types of organosulfur compounds, hydrolysis in the liquid phase after capture into the instrument and before analysis might lead to SAGA-MC detecting these as well, hence  the SAGA-MC sulfate measurements based on the default accuracy estimates for this instrument for inorganic sulfate are likely more uncertain under FIREX-AQ conditions (Dibb et al., 2002; Scheuer et al., 2003).

Both IC (SAGA) instruments detect HMS as S(IV), and the signal interfered with sulfite and bisulfite. There is no unambiguous detection of HMS specifically, either in the IC or in the AMS.

The following text has been added to the discussion of SAGA-MC detection of S(IV):

However, the S(IV) from the SAGA-MC is comparable to the SAGA filter samples, which are unaffected by ambient $SO_2$ and hence suggests that a large fraction of the S(IV) in the SAGA-MC was present in the aerosol, and that the contribution of the $SO_2$ artifact to the S(IV) signal is small. This observation further suggests that most of the S(IV) was present in submicron particles, as supermicron particles are not quantified by the SAGA-MC (Guo et al, 2021).

Follow-up comment #3

Initial Comment #6: Lines 347 to 362: Maybe the lack of correlation with MCE is that the MCE dynamic range is small? One might try looking at BC/OA ratios, just out of curiosity.

Reply to Comment: Because the range of MCE values presented in Fig. 3 covers a similar range as used in the cited analyses, we believe this range is adequate for the determination of a lack of correlation.

New Comment: In other studies (in fact analysis of this FIREX data), it has been shown that BC/OA is in fact better than MCE for separating out some properties that depend on smoldering vs flaming, I suggest the authors do the calculation to actually test it.

While different analysis might reveal a relationship between sulfur emission factors and a different metric for fire stage, we focus here on MCE because of the historical president and widespread use of this parameter, and for comparison with the studies already cited in this paper. Additional analysis is beyond the scope of this work.

Follow-up comment #4

Initial Comment #10: Why not make plots of pH and LWC vs plume age for all these modeled plumes, given that these are key variables?

Reply to Comment: We believe that the trends of these variables are clear in the data presented in the supplementary materials.

New Comment: The trends are not clear. For example, I cannot tell from Fig S3 what causes the systematic jump in pH back and forth between two levels, is this due to in vs out of plumes (it is hard to line up with the SO2 and sulfate data. The pH does seem to increase when the aircraft climbs. Same applies to Fig S4, expect in this case the plot shows no pH when the aircraft changes altitude. Some form of plot other than a time series I think would help clarify the odd behavior in pH.

This is correct that the jumps in pH are due to entering and exiting the plume. The peaks are representative of the center of the transect and the troughs are representative of outside of the plume. The values increase and decrease upon entering and exiting the plume, respectively. The plume transects have now been highlighted in Fig. S3 for clarification. A plot that is being described could be shown; however, this would only be representative of the two flights that were modeled. There are, in fact, more data for the remainder of the flights; however, it would take considerable time to analyze each transect for pH, LWC, and plume age which at this time is not feasible. A clear pH shift with altitude was only observed (of the modeled flights) for the 7 August flight as the reviewer has noted. This was likely due to the chemical compounds present and the decreased temperature observed with the changing altitude as well as potentially changing meteorological conditions. With the available data from the two modeled flights, it is believed that a plot of this type with limited data would not add significant insight to the study.

Follow-up comment #5

I would like to thank the authors for addressing the comments. I have a minor comment regarding the units of LWC in lines 638-639. The units are provided in g sm-3 but I assume that they are meant to be g cm-3. Please check and correct accordingly.

The units reported in the paper are correct. The LWC values are reported in units of grams per standard cubic meter. These units have now been defined on line 746.

Follow-up comment #6

Following up on the previous Comment #5 (regarding the HMS chemistry and the effect of hydroxymethyl hydroperoxide formation) the authors added a sentence stating the potential effect of this chemistry, however a quick test with the model of an average scenario would be useful. I would recommend such a test to be added in the SI with the sentence already presented in the main text in order to have an estimate of the magnitude of hydroxymethyl hydroperoxide chemistry in their system.

Because the main focus of this modeling section is to understand the influence of HMS chemistry on the evolution of SO2 to sulfate, the authors leave the suggestion of hydroxymethyl hydroperoxide chemistry for future studies to consider.